# p85β regulates autophagic degradation of AXL to activate oncogenic signaling

Ling Rao [1,7], Victor C. Y. Mak[1,7], Yuan Zhou[1], Dong Zhang[2], Xinran Li[1], Chloe C. Y. Fung[1], Rakesh Sharma[3], Chao Gu[4], Yiling Lu[5], George L. Tipoe[1], Annie N. Y. Cheung[6], Gordon B. Mills [2] & Lydia W. T. Cheung [1✉]

*PIK3R2* encodes the p85β regulatory subunit of phosphatidylinositol 3-kinase and is frequently amplified in cancers. The signaling mechanism and therapeutic implication of p85β are poorly understood. Here we report that p85β upregulates the protein level of the receptor tyrosine kinase AXL to induce oncogenic signaling in ovarian cancer. p85β activates p110 activity and AKT-independent PDK1/SGK3 signaling to promote tumorigenic phenotypes, which are all abolished upon inhibition of AXL. At the molecular level, p85β alters the phosphorylation of TRIM2 (an E3 ligase) and optineurin (an autophagy receptor), which mediate the selective regulation of AXL by p85β, thereby disrupting the autophagic degradation of the AXL protein. Therapeutically, p85β expression renders ovarian cancer cells vulnerable to inhibitors of AXL, p110, or PDK1. Conversely, p85β-depleted cells are less sensitive to these inhibitors. Together, our findings provide a rationale for pharmacological blockade of the AXL signaling axis in *PIK3R2*-amplified ovarian cancer.

[1] School of Biomedical Sciences, Li Ka Shing Faculty of Medicine, The University of Hong Kong, Hong Kong, Hong Kong. [2] Knight Cancer Institute, Oregon Health and Science University, Portland, Oregon, USA. [3] Proteomics and Metabolomics Core Facility, Li Ka Shing Faculty of Medicine, The University of Hong Kong, Hong Kong, Hong Kong. [4] Department of Obstetrics and Gynecology, Obstetrics and Gynecology Hospital, Fudan University, Shanghai, China. [5] Department of Systems Biology, The University of Texas MD Anderson Cancer Center, Houston, Texas, USA. [6] Department of Pathology, Li Ka Shing Faculty of Medicine, The University of Hong Kong, Hong Kong, Hong Kong. [7] These authors contributed equally: Ling Rao, Victor C. Y. Mak. ✉email: lydiacwt@hku.hk

The p85 regulatory subunit of class 1A phosphatidylinositol 3-kinase (PI3K) binds to the p110 catalytic subunit, initiating signaling downstream of activated receptor tyrosine kinases (RTKs). Although there is functional specificity among p110 isoforms[1,2], the roles of p85 isoforms in cancers are only beginning to be understood. *PIK3R1* (encodes p85α) has been suggested to act as a tumor suppressor through functions such as inhibiting p110 kinase activity and stabilizing phosphatase and tensin homolog (PTEN)[3,4]. Depletion of p85 can thus lead to enhanced p110 activity and PTEN destabilization, as well as cell context-dependent activation of oncogenic signaling[3–5]. Indeed, loss-of-function disruptions in *PIK3R1* are frequent in cancers, including copy number loss and truncation or point mutations. In contrast, mutations in *PIK3R2* (p85β) are uncommon, with gene amplification being observed more often than mutations. Concordant with the *PIK3R2* genomic profile, we and others have demonstrated that the expression of p85β confers tumorigenic properties. Phenotypic studies using cancer models have demonstrated that *PIK3R2* depletion decreases the viability of a breast cancer cell line in vitro and hampers colon carcinogenesis in *Pik3r2*[−/−] mice[6]. p85β also promotes invadopodium formation and melanoma invasion, possibly through increased activity of the GTPases Cdc42 and Rac1[7]. Further, ectopic expression of p85α inhibits AKT activation in endometrial cancer cells, but such inhibition is not observed in p85β-expressing cells[8]. Although these data collectively implicate *PIK3R2* as an oncogene, the downstream signaling events and associated activating mechanisms selectively triggered by *PIK3R2* have yet to be elucidated.

Here we report that p85β signals through its upstream kinase AXL, which in turn activates p110 to induce PDK1/SGK3 signaling, establishing the mechanistic basis for targeting AXL in *PIK3R2*-amplified ovarian cancer. We also reveal a p110 activity-independent role of p85β in regulating the autophagy-lysosomal machinery and thereby affecting AXL protein degradation.

## Results

**p85β upregulates the activity but not the expression of p110.** An increased *PIK3R2* copy number was detected in 49% of The Cancer Genome Atlas (TCGA) serous ovarian tumor samples ($n = 579$). *PIK3R2* copy numbers positively correlated with corresponding mRNA levels measured by RNA-Sequencing ($r = 0.7$, $P < 0.0001$) or microarray ($r = 0.6$, $P < 0.0001$) (Supplementary Fig. 1a). *PIK3R2* mRNA expression was higher in *PIK3R2*-amplified tumors than in diploid tumors ($P < 0.0001$) (Supplementary Fig. 1b). Intriguingly, high *PIK3R2* mRNA levels were significantly associated with relatively poor overall survival and progression-free survival in ovarian cancer patients (Fig. 1a).

The functional consequences of silencing *PIK3R2* were evaluated in three serous ovarian cancer cell lines with high p85β protein levels (OVCAR4, OVCAR8, and SKOV3) using two independent small interfering RNA (siRNA). Knockdown efficiency is shown in Supplementary Fig. 1c. *PIK3R2* depletion impaired cell proliferation, long-term clonogenic survival, and cell invasion (Fig. 1b–d). Stable short hairpin RNA (shRNA)-mediated *PIK3R2* knockdown induced similar phenotypic changes in vitro and decreased intraperitoneal growth in vivo (Supplementary Fig. 1c–g). To evaluate the functional consequences of increased p85β levels, p85β was stably expressed in serous ovarian cancer cell lines with low endogenous p85β protein levels (DOV13 and EFO21). This p85β overexpression led to enhancements in tumorigenic phenotypes (Fig. 1e–g and Supplementary Fig. 2a–c). These increases were markedly abolished by pan-p110 inhibitors (GDC-0941; PIK-90), p110α-specific inhibitors (A66; BYL719), or

a p110β-specific inhibitor (TGX-221), indicating the contribution of p110 to the activity of p85β (Fig. 1e–g and Supplementary Fig. 2a–c). Remarkably, two AKT inhibitors (MK-2206; GDC-0068) did not alter the induced phenotypes, indicating that the effects of p85β are independent of AKT signaling. This is further supported by the observation that knocking down AKT1/2/3 expression with siRNA had minimal impacts on the p85β-induced phenotypes (Supplementary Fig. 2d).

p85α binds to p110 to stabilize p110 proteins and inhibit p110 kinase activity[9]. Strikingly, we found that p85β promoted p110 kinase activity, which was reflected by the production of phosphatidylinositol 3,4,5-trisphosphate (PIP3) from phosphatidylinositol 4,5-bisphosphate (PIP2) in p110α- or p110β-immunoprecipitated lysates (Fig. 1h). p110 activity associated with either p85α or p85β was enhanced (Supplementary Fig. 2e); therefore, the effect is not limited to p85β-bound p110. Total cellular PIP3 levels in p85β-overexpressing cells were also increased (Supplementary Fig. 2f). Consistently, p110 activities and PIP3 levels were decreased by *PIK3R2* siRNA (Supplementary Fig. 2g, h). However, the protein levels of p110α and p110β were not altered in p85β-overexpressing (Fig. 1i) or p85β-depleted cells (Supplementary Fig. 2i).

**p85β increases the protein level of AXL to activate p110.** To understand the effects of p85β, we analyzed TCGA serous ovarian cancer protein data generated by reverse-phase protein array (RPPA). The samples were divided according to *PIK3R2* copy number. Intriguingly, the levels of AXL (an RTK in the TAM family) were significantly elevated in the tumors with *PIK3R2* amplification ($P = 0.01$). We also demonstrated a positive correlation between p85β and AXL levels in an independent serous ovarian tumor set ($n = 49$) by immunohistochemistry (Pearson's correlation coefficient $r = 0.6$, $P < 0.0001$; Fig. 2a, b). Western blotting confirmed that both the AXL protein level (Fig. 2c) and tyrosine phosphorylation (Fig. 2d) were increased in p85β-overexpressing DOV13 and EFO21 cells. Conversely, *PIK3R2* siRNA treatment or stable *PIK3R2* knockdown led to decreases in the AXL protein level and phosphorylation (Fig. 2e, f and Supplementary Fig. 3a). Remarkably, depletion of p85α (*PIK3R1*) had no appreciable effect on the AXL protein level (Supplementary Fig. 3b), indicating the specific effect of *PIK3R2*. Further, this regulation was specific to AXL, because the protein levels of other TAM members (MERTK and Tyro3) were not changed (Supplementary Fig. 3c).

As AXL is an upstream regulator of PI3K, we investigated the involvement of AXL in p85β-induced phenotypes and p110 activity. Cell proliferation, colony formation, and cell invasion induced by p85β were all abrogated upon inhibition of AXL by dominant-negative (DN) AXL or an AXL inhibitor (TP-0903) (Fig. 2g–i and Supplementary Fig. 3d–f). The induced catalytic activities of p110α and p110β were also abolished by AXL inhibition (Fig. 2j and Supplementary Fig. 3g). Together, these data indicate that p85β increases the AXL protein level and activation to promote p110 activities, which may in turn initiate downstream oncogenic signaling.

The regulation of AXL by p85β was, however, independent of p110 activity. Treatment with the pan-110 inhibitor (PIK-90), p110α-specific inhibitor (BYL719), or p110β-specific inhibitor (TGX-221) did not abolish the increased AXL levels by p85β in two cell lines (Fig. 3a and Supplementary Fig. 4a). Phosphorylation of AKT and S6 was inhibited reflecting the effectiveness of the inhibitors. Crosstalk between AXL and EGFR family members, for example, via AXL-EGFR heterodimerization, has previously been demonstrated[10,11]. Neither lapatinib (targets EGFR and HER2) nor erlotinib (targets EGFR) had any influence

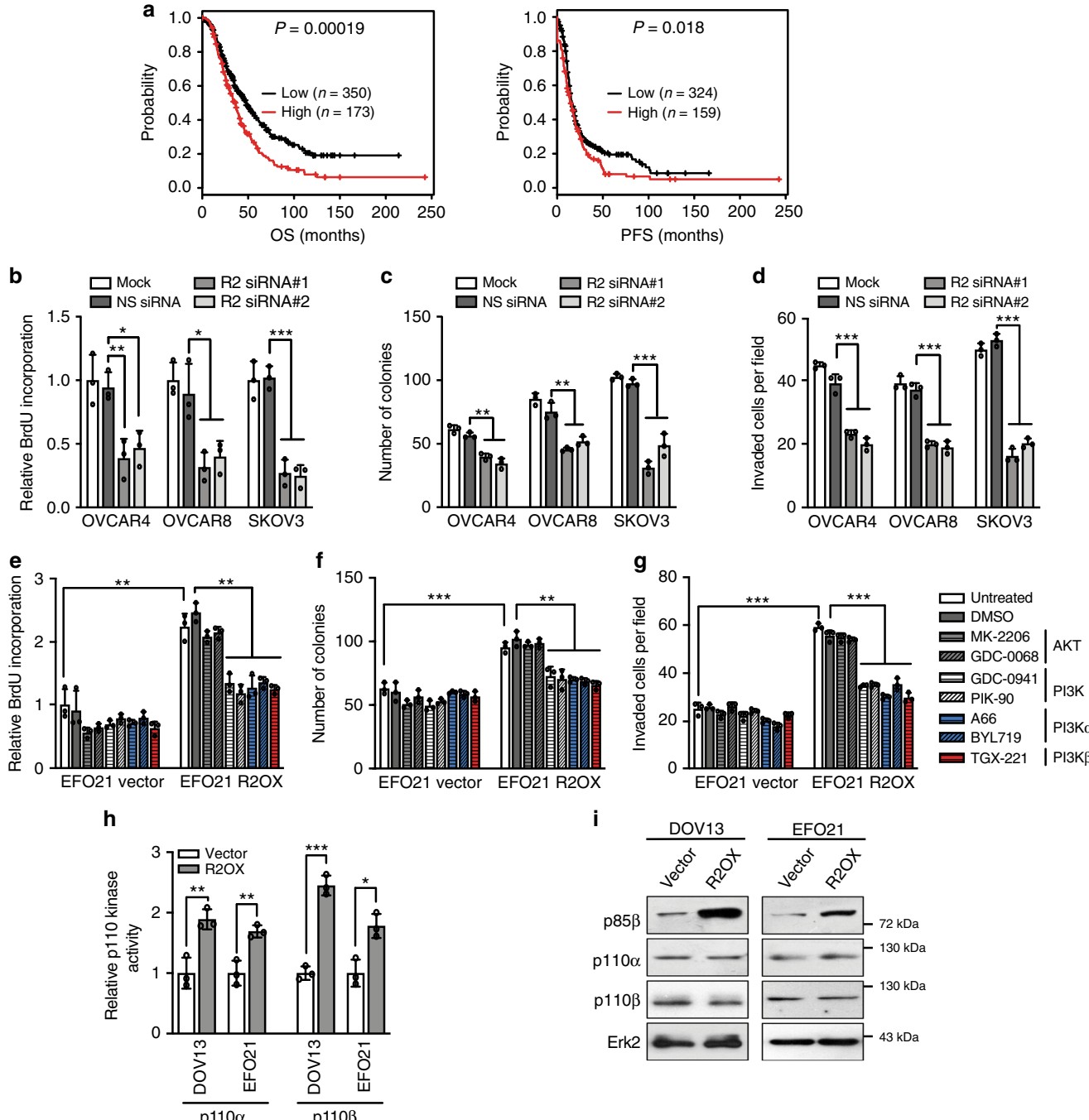

**Fig. 1 Oncogenicity of p85β depends on p110 activities but is independent of AKT. a** Overall survival (OS) and progression-free survival (PFS) of serous ovarian cancer patients split at the upper tertile of *PIK3R2* mRNA level. Data were obtained from Kaplan–Meier Plotter using both GEO and TCGA datasets. Two-sided logrank test *P*-values of the two-group comparison are shown. **b–d** Ovarian cancer cells with or without *PIK3R2* silencing (R2 siRNA) were examined for (**b**) BrdU cell proliferation, (**c**) colony formation, and (**d**) cell invasion. NS siRNA, nonspecific siRNA. **e–g** EFO21 cells stably expressing *PIK3R2* (R2OX) or empty vector were treated with the indicated inhibitors and subjected to (**e**) BrdU cell proliferation assay, (**f**) colony formation assay, and (**g**) cell invasion assay. **h** p110α or p110β proteins were immunoprecipitated from protein lysates of cells with or without stable *PIK3R2* overexpression. The eluants were subjected to PI3-kinase activity assay. **i** Protein levels of p85β, p110α, p110β, and Erk2 (a loading control) were examined by western blotting. The western blotting experiment was repeated three times with independent lysates and results were reproducible. Assays in **b–h** were done in triplicate. Data shown are representative of three independent experiments and presented as mean ± SD. *P < 0.05; **P < 0.01; ***P < 0.001 using two-tailed *t*-test. Source data are provided as a Source Data file.

on the increased phosphorylation and total protein level of AXL induced by p85β (Supplementary Fig. 4b), suggesting that the activation of AXL is unlikely to occur through an interaction with EGFR or HER2. Gas6 is a ligand that activates AXL but may also decrease the AXL level by promoting AXL protein degradation[12]. Our data showed that the level of Gas6 was unaffected by *PIK3R2* depletion (Supplementary Fig. 4c), and that Gas6 had minimal interference with the effect of p85β on AXL. First, Gas6 caused

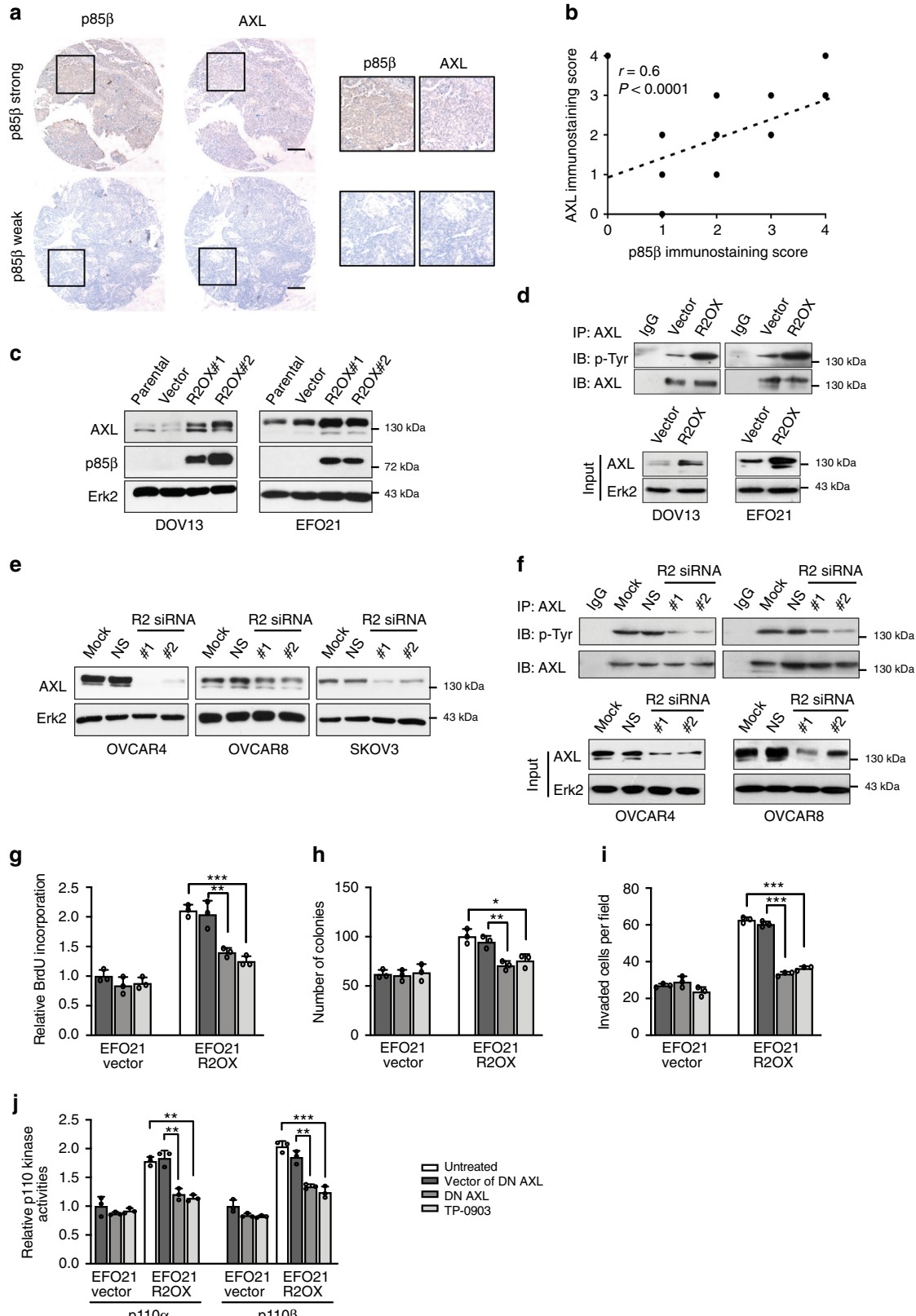

transient activation of AXL in vector control-expressing or p85β-overexpressing cells (Fig. 3b). Importantly, the induced phosphorylation of AXL by p85β and Gas6 was additive. Second, total AXL protein levels remained high in p85β-overexpressing cells compared with vector control-expressing cells after Gas6 stimulation (Fig. 3b). Third, we examined the potential

effect of Gas6 on *PIK3R2*-knockdown cells. The decreases in AXL phosphorylation and protein level upon *PIK3R2* depletion were not affected by Gas6 (Fig. 3c). Last, the protein interaction between p85β and AXL remained essentially unchanged in Gas6-stimulated cells, as revealed by an in situ proximity ligation assay (PLA) (Fig. 3d).

**Fig. 2 p85β increases the protein level of AXL, which in turn activates p110. a** Serous ovarian tumor tissue samples were subjected to immunohistochemical staining for p85β and AXL. Scale bars, 200 μm. The boxes depict magnified areas. **b** Scatter plot illustrating the correlation of the immunostaining scores corresponding to p85β and AXL. Pearson's correlation coefficient *r* with *P*-value are shown. **c** Protein levels of AXL, p85β, and Erk2 (loading control) in cells stably expressing *PIK3R2* (R2OX) or vector control. **d** AXL was immunoprecipitated (IP) from whole cell lysates and western blotting (IB) was performed with anti-p-Tyr and anti-AXL antibodies (upper). IP with mouse IgG was negative control. AXL levels in the input lysates are shown (lower). AXL protein levels were normalized prior to IP by using proportionally different amounts of input lysates. **e** Protein levels of AXL and p85β in cells with or without *PIK3R2* silencing. NS, nonspecific siRNA. **f** AXL was immunoprecipitated from cell lysates and western blotting was performed (upper). AXL levels in input lysates is shown (lower). **g–j** EFO21 cells with or without stable *PIK3R2* overexpression were either treated with AXL inhibitor TP-0903 (0.5 μM) or transfected with dominant-negative (DN) AXL or its vector control. These cells were subjected to (**g**) BrdU cell proliferation assay, (**h**) colony formation assay, (**i**) cell invasion assay, and (**j**) immunoprecipitation-based p110α or p110β PI3-kinase activity assay. The western blotting experiments (**c–f**) were repeated three times with independent lysates and results were reproducible. Assays in **g–j** were done in triplicate and representative data of three independent experiments are presented as mean ± SD. *\*P* < 0.05; *\*\*P* < 0.01; *\*\*\*P* < 0.001 using two-tailed *t*-test. Source data are provided as a Source Data file.

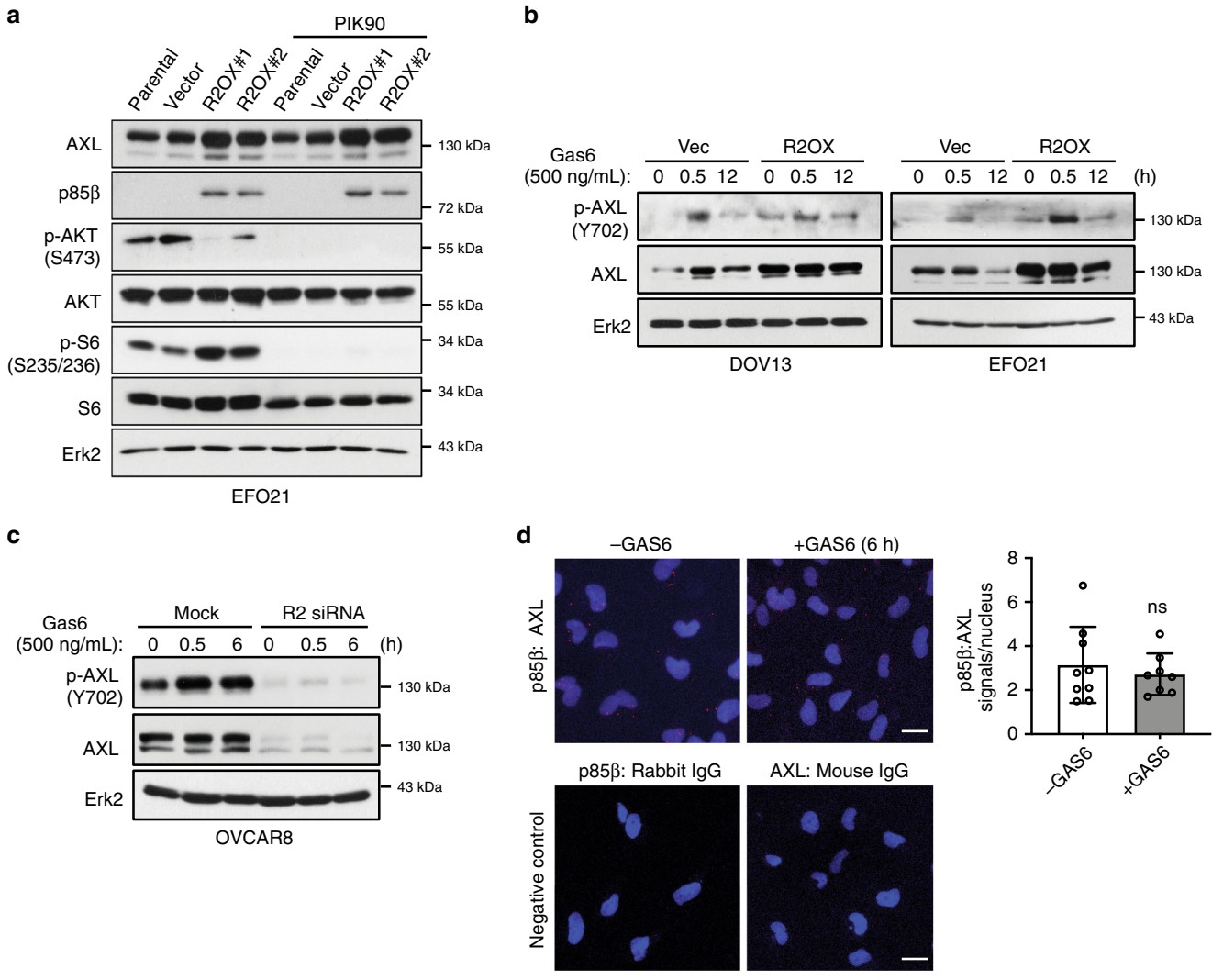

**Fig. 3 The regulation of AXL by p85β does not involve p110 activity and is not affected by Gas6. a** EFO21 cells with or without stable *PIK3R2* overexpression (R2OX) were treated with 10 μM PIK-90 (pan-p110 inhibitor) for 24 h. Protein was then collected and analyzed using the indicated antibodies. **b, c** DOV13 or EFO21 cells stably expressing *PIK3R2* (R2OX) or vector control or (**c**) OVCAR8 cells with or without *PIK3R2* silencing (R2 siRNA) were serum-starved for 24 h. Cells were then treated with human recombinant Gas6 (500 ng/mL) for the indicated times. The western blotting experiments were repeated three times with independent lysates and results were reproducible. **d** Parental OVCAR8 cells were serum-starved for 24 h prior to stimulation with Gas6 (500 ng/mL) for 6 h. Cells were subjected to proximity ligation assay with anti-p85β and anti-AXL antibodies. Negative controls with one of the antibodies replaced by IgG were included. The assay was repeated three times and reproducible results were obtained. Representative images are shown (left). The number of signals per nucleus was counted in ≥8 fields and data represent mean ± SD (right). Scale bars, 20 μm. ns, not statistically significant by two-tailed *t*-test. Source data are provided as a Source Data file.

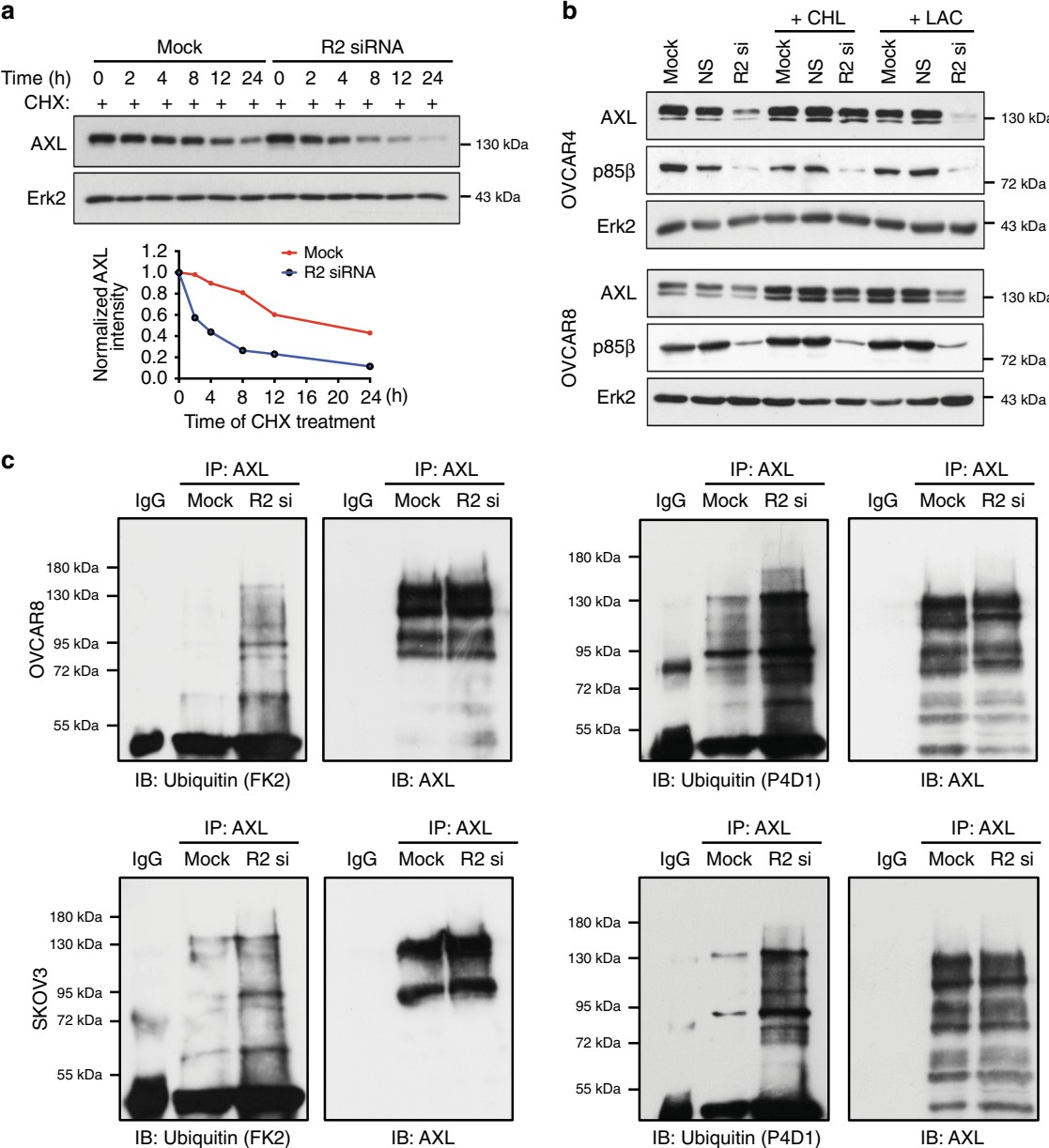

**Fig. 4 p85β stabilizes AXL via decreasing AXL ubiquitination. a** SKOV3 cells transfected with *PIK3R2* siRNA for 36 h were then treated with 10 μg/mL cycloheximide (CHX, inhibitor of protein synthesis) for the indicated time course. Protein levels of AXL were examined by western blotting and quantified by densitometry using Image J, with Erk2 as a loading control. **b** Cells transfected with *PIK3R2* siRNA (R2 si) for 36 h were treated with 100 μM chloroquine (CHL, lysosomal inhibitor) or 5 μM Lactacystin (LAC, proteasomal inhibitor) for another 24 h. NS, nonspecific siRNA. **c** AXL was immunoprecipitated (IP) from lysates of cells transfected with or without *PIK3R2* siRNA for 72 h. AXL protein levels were normalized prior to IP by using proportionally different amounts of input lysates. Independent sets of samples were subjected to western blotting (IB) using anti-Ubiquitin antibodies (clones FK2 and P4D1). The membranes were stripped and re-probed with anti-AXL antibody. IP with mouse IgG was negative control. The experiments were repeated three times with independent lysates and results were reproducible. Source data are provided as a Source Data file.

**p85β regulates the autophagic degradation of AXL.** Although p85α has been implicated in the recycling of activated platelet-derived growth factor receptor in NIH3T3 cells[13], the regulation of any RTK by p85β has not been reported. The *AXL* mRNA level and promoter activity were not changed in *PIK3R2*-depleted cells (Supplementary Fig. 5a–b). Cycloheximide-chase experiments revealed that *PIK3R2* depletion reduced the half-life of the endogenous AXL protein (Fig. 4a and Supplementary Fig. 5c). Proteins are mainly degraded by the proteasomal or lysosomal pathway. As shown in Fig. 4b and Supplementary Fig. 5d, *PIK3R2* depletion failed to decrease the AXL protein level upon treatment with chloroquine (lysosome inhibitor), but not with MG132 or

lactacystin (proteasome inhibitors), suggesting that AXL is regulated by p85β via the lysosomal pathway. Ubiquitination was analyzed by immunoblotting with two different ubiquitin-specific antibodies prior to stripping the membranes for reprobing with anti-AXL antibody. The anti-ubiquitin clone FK2 recognizes both conjugated monoubiquitin and polyubiquitin chains, whereas the clone P4D1 recognizes all forms of ubiquitin including conjugated or free ubiquitin. Notably, our data revealed an increase in ubiquitinated AXL in the presence of *PIK3R2* siRNA (Fig. 4c). In addition to distinct bands that may correspond to mono-ubiquitinated AXL, we observed faint high-molecular-weight smears in *PIK3R2*-depleted samples under prolonged blot

exposures (Supplementary Fig. 5e). Precise characterization of the types of ubiquitination is warranted.

We next attempted to identify the proteins that mediate the entry of AXL into the degradation pathway. Phosphorylation modulates the activity and substrate selectivity of proteins within the degradation machinery[14,15]. Quantitative mass spectrometry (MS)-based phosphoproteomic analysis was therefore performed to reveal proteins with an altered phosphorylation status upon *PIK3R2* depletion. This profiling yielded 90 unique altered phosphopeptides, mapping to 29 and 45 proteins whose levels were downregulated or upregulated at $P < 0.05$ by *PIK3R2*-specific siRNA, respectively (Supplementary Table 1). These proteins were analyzed by Gene Ontology (GO) enrichment analysis. Strikingly, we observed significant enrichment in ubiquitin-mediated proteolysis (log10 *P*-value = −5.36) and autophagy regulation (log10 *P*-value = −2.58) (Fig. 5a). Selective autophagy represents a major lysosome-mediated degradation mechanism in which ubiquitinated proteins are recognized by autophagic receptors forming autophagosomes prior to fusion with the lysosome[16,17].

We performed siRNA experiments targeting the 12 proteins with phosphorylation changes within these two GO categories (Supplementary Fig. 6a). In addition, we included siRNAs targeting the E3 ubiquitin ligases Cbl-b and c-Cbl, because these ligases have been demonstrated to mediate the degradation of RTKs including AXL[12,18]. The screen revealed that knocking down the expression of two genes, *OPTN* and *TRIM2*, could reverse the decrease in the AXL protein level induced by *PIK3R2* depletion. In contrast, the inhibition of the other proteins, including Cbl-b or c-Cbl, had no effect (Supplementary Fig. 6a). The involvement of *TRIM2* and *OPTN* in AXL degradation was further confirmed by multiple independent siRNAs (Fig. 5b). NEDD4, which has been shown to be an RTK E3 ligase[19,20], was chosen as a negative control, because it had no effect on AXL levels in the screen. Consistently, *TRIM2* and *OPTN* knockdown but not *NEDD4* knockdown reversed *PIK3R2* loss-induced AXL degradation (Fig. 5b). Remarkably, *TRIM2*- and *OPTN*-specific siRNAs had no effect on Tyro3 or MERTK, indicating the specificity of their activities (Supplementary Fig. 6b).

TRIM2 is a tripartite motif (TRIM) RING finger E3 ligase that ubiquitinates the neurofilament light subunit[21] and the pro-apoptotic protein Bim[22]. Our data suggested TRIM2 as an E3 ligase of AXL. Depletion of TRIM2 by siRNA reduced the amount of ubiquitinated AXL (Fig. 5c), whereas TRIM2 overexpression reversed the upregulation of the AXL level mediated by p85β (Fig. 5d). Importantly, phosphorylation-defective mutation of S443 in TRIM2 (TRIM2^S443A), a residue whose phosphorylation was increased by p85β knockdown, rendered TRIM2 incapable of downregulating AXL level. S443 is located within the filamin domain in TRIM2, but the function of this domain is unknown. Binding between AXL and TRIM2 was promoted by *PIK3R2* knockdown, as revealed by immunoprecipitation (Fig. 5e) and a PLA (Supplementary Fig. 6c). Importantly, Gas6 stimulation at a condition that effectively induced AXL phosphorylation did not disrupt the binding (Supplementary Fig. 6c).

*OPTN* encodes optineurin, which is a selective autophagy receptor that mediates the interactions between ubiquitinated proteins and the autophagy machinery. Immunoprecipitation and PLA consistently demonstrated that *PIK3R2* knockdown enhanced the binding between AXL and optineurin (Fig. 5e), which was again unaffected by Gas6 (Supplementary Fig. 6d). *PIK3R2* knockdown increased the phosphorylation of optineurin at S526, which is a highly conserved residue close to the ubiquitin-binding domain that has not been previously reported to be phosphorylated. We confirmed the involvement of optineurin in AXL degradation, because the increased AXL level

induced by p85β was abrogated upon optineurin overexpression (Fig. 5f). The phosphorylation-defective optineurin mutant S526A failed to decrease the AXL level (Fig. 5f), suggesting the importance of optineurin phosphorylation in the degradation of AXL. We also investigated the involvement of the other previously characterized amino acids in optineurin, including S177, S473, and S513, which may alter ubiquitin binding or other functions of optineurin in mitophagy[23–27]. S473A but not S177A or S513A reversed the decrease in the AXL level induced by p85β (Supplementary Fig. 6e). Double mutants composed of S526A with S177A or S473A had effects similar to those of single S526A mutant. Western blotting with a commercial antibody against S177 showed that its phosphorylation was unaffected by p85β (Supplementary Fig. 6f).

Autophagic flux, which reflects autophagic degradation activity, was induced by *PIK3R2* depletion. The protein level of the autophagosome marker LC3B-II was increased by *PIK3R2* depletion and treatment with chloroquine, which inhibits autophagosome–lysosome fusion[28], resulted in further accumulation of LC3B-II (Fig. 5g). This increase in LC3B-II level was accompanied by an increase in LC3B puncta formation in *PIK3R2*-depleted cells, as revealed by confocal double-labeling immunofluorescence (Fig. 5h). Further, we observed pronounced colocalization between AXL and LC3B, ATG5 and ATG12 (autophagy markers)[29,30], or Rab7 (a Rab GTPase that regulates late endosomal trafficking and the autolysosomal pathway[31] in *PIK3R2*-depleted cells (Fig. 5h)). The presence of AXL in these vesicles is consistent with degradation through the autophagy-lysosomal pathway. In contrast, the associations between AXL and Rab4 or Rab11 (which regulated receptor recycling to the plasma membrane[31] were not altered by *PIK3R2* depletion (Supplementary Fig. 6g).

**p85β signals through SGK3 instead of AKT**. To elucidate the downstream signaling events that mediate the effects of p85β, five ovarian cancer cell lines transfected with *PIK3R2*-specific siRNA were subjected to RPPA analysis. AXL level was downregulated in four of the five cell lines (Fig. 6a). Consistent with the observation above (Fig. 5g), we observed increased LC3 protein levels in *PIK3R2*-knockdown cells. Intriguingly, *PIK3R2*-specific siRNA also led to decreased amounts of p-PDK1 (S241), an upstream kinase of AKT and SGK[32], and p-NDRG1 (pT346), a marker of SGK activity[33,34] (Fig. 6a). Consistent with the observations that the effects of *PIK3R2* were insensitive to AKT inhibitors, an RPPA and subsequent western blotting validation showed no significant differences in the levels of phosphorylated AKT, the AKT substrate PRAS40, mammalian target of rapamycin (mTOR), p70S6K, and S6 in *PIK3R2*-depleted cells (Fig. 6b). Indeed, AKT phosphorylation at S473 was modestly reduced in *PIK3R2*-overexpressing cells (Fig. 3a and Supplementary Fig. 7a). Phosphorylation of PRAS40, mTOR, and p70S6K was not changed in these *PIK3R2*-overexpressing cells (Supplementary Fig. 7a). SGK has been proposed as to be a downstream effector of AKT-independent PI3K oncogenic signaling[35,36]. SGK antibodies were not included in the RPPA due to inability to validate on that platform. We therefore evaluated the phosphorylation of SGK1 and SGK3 using western blotting. Although the p-SGK1 (S78) level was not altered, the p-SGK3 (T320) level was decreased upon *PIK3R2* silencing (Fig. 6b). The decreases in the p-PDK1 and p-NDRG1 levels were reproducible across the cell lines (Fig. 6b). Importantly, p85β-overexpressing ovarian cancer cells demonstrated coordinated enhanced phosphorylation of PDK1, SGK3, and NDRG1 (Fig. 6c).

Next, we sought to assess the contribution of each of these molecules to the effects of p85β on SGK3 signaling by inhibiting

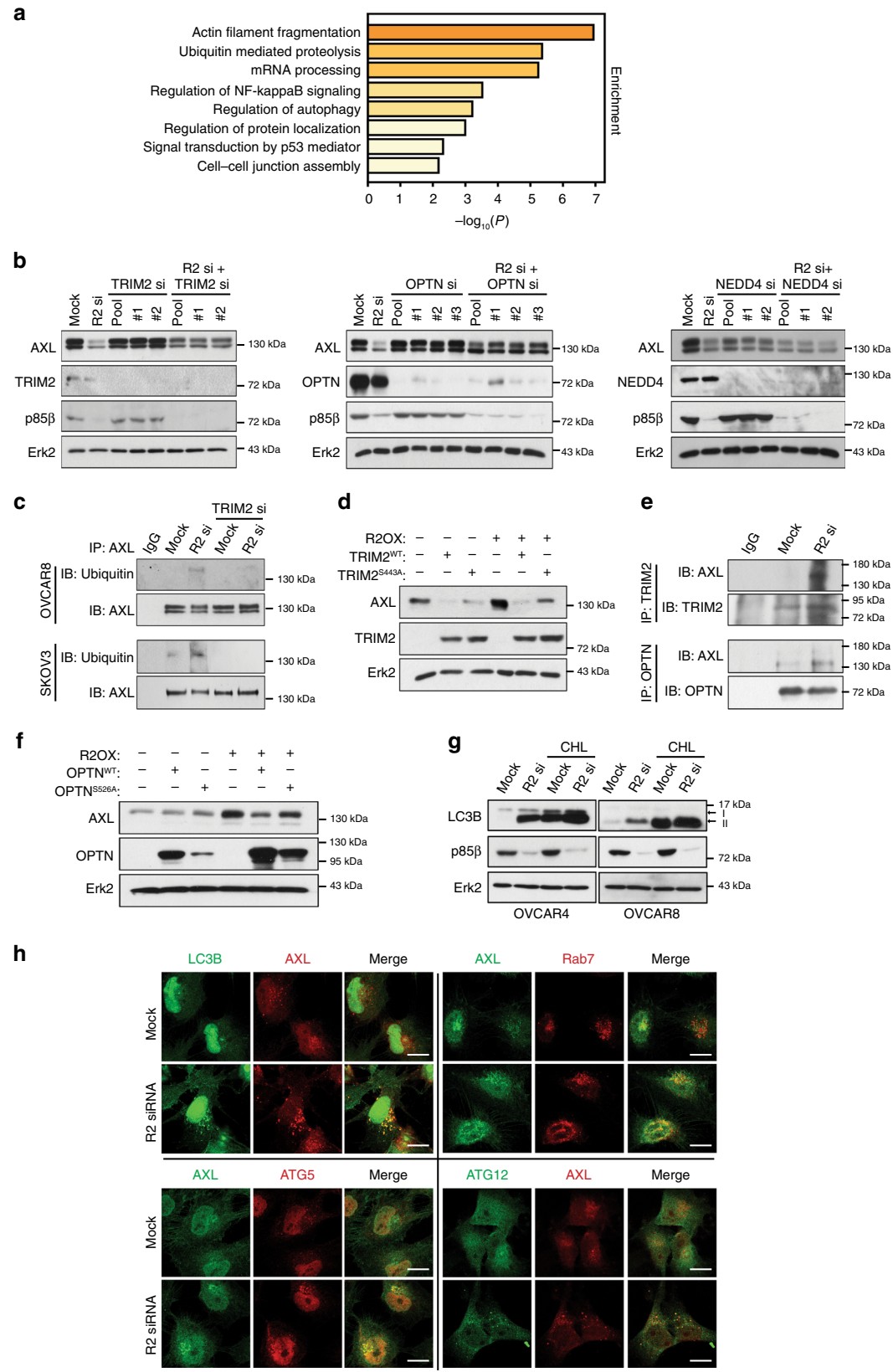

AXL (TP-0903), p110 (GDC-0941), or PDK1 (GSK2334470). On-target inhibition of AXL phosphorylation and p110 activity by TP-0903 or GDC-0941 is shown in Fig. 6c and Supplementary Fig. 7b, respectively. Inhibition of PDK1 phosphorylation could also be observed upon GSK2334470 treatment (Fig. 6c). All the

inhibitors suppressed the induction of PDK1, SGK3, and NDRG1 phosphorylation by p85β expression (Fig. 6c). These results collectively indicate activation of the AXL/p110/PDK1/SGK3/NDRG1 signaling cascade by p85β in ovarian cancer. Notably, these inhibitions had no effect on the induction of AXL protein

**Fig. 5 TRIM2 and optineurin mediate the autophagy-lysosomal degradation of AXL regulated by p85β. a** Gene ontology enrichment analysis was performed with proteins of which phosphorylation was significantly altered upon *PIK3R2* silencing in SKOV3 cells. **b** OVCAR8 cells were transfected with *PIK3R2* siRNA, alone or in combination with siRNA targeting *TRIM2*, *OPTN*, or *NEDD4* for 72 h. Protein levels of AXL, TRIM2, OPTN, NEDD4, p85β, and Erk2 (loading control) were examined. **c** Cells were transfected with *PIK3R2* siRNA, with or without *TRIM2* siRNA for 72 h. Cell lysates were immunoprecipitated with anti-AXL antibody and subsequently analyzed using anti-Ubiquitin and anti-AXL antibodies. **d** EFO21 cells stably expressing *PIK3R2* (R2OX) or vector were transfected with expression plasmids of *TRIM2* or mutant (*TRIM2* S443A) for 72 h. Protein levels of AXL, TRIM2, and Erk2 was examined. **e** OVCAR8 cells were transfected with *PIK3R2* siRNA for 72 h. Cell lysates were immunoprecipitated (IP) with anti-TRIM2 or anti-OPTN antibody followed by western blotting (IB). **f** EFO21 cells stably expressing *PIK3R2* (R2OX) or vector were transfected with expression plasmids of *OPTN* or mutant (*OPTN* S526A) for 72 h. Protein levels of AXL, OPTN, and Erk2 were examined. **g** Cells transfected with siRNA for 48 h were treated with 100 μM chloroquine for another 12 h. Protein levels of autophagic marker LC3B, p85β, and Erk2 were examined. **h** OVCAR8 cells transfected with siRNA for 48 h were fixed and subjected to immunofluorescence staining using antibodies against AXL, LC3B, Rab7, and autophagy marker ATG5 or ATG12. Scale bars, 20 μm. The experiments were repeated three times with independent samples and results were reproducible. Source data are provided as a Source Data file.

levels by p85β (Fig. 6c), concordant with the notion that the regulation of AXL by p85β is independent of p110 activity. To further investigate whether this signaling axis in turn activates p110 activity, we assessed PIP3 production in p110α or p110β immunoprecipitates from *PIK3R2*-overexpressing cells treated with PDK1 inhibitors (GSK2334470 and OSU-03012). The inhibition of PDK1 did not abolish p85β-induced p110α or p110β activity, consistent with PDK1 being downstream of p110 (Supplementary Fig. 7c).

**p85β sensitizes cells to inhibition of AXL signaling**. p85β could confer sensitivity to inhibitors targeting molecules in the AXL/p110/PDK1/SGK3/NDRG1 signaling cascade. p85β-overexpressing 3D spheroids were treated with serial dilutions of inhibitors of AXL and PI3K/PDK1 signaling to obtain IC50 values in the presence and absence of p85β expression. Strikingly, p85β-overexpressing 3D spheroids displayed significantly higher sensitivity to AXL (BGB324 and TP-0903), pan-PI3K (GDC-0941), p110α-specific (A66), p110β-specific (TGX-221), and PDK1 (OSU-03012 and GSK2334470) inhibitors than control spheroids (Fig. 7a, c). Drug sensitivity was also assessed in spheroids of *PIK3R2*-depleted cells. In contrast to the results for the overexpression spheroids, *PIK3R2* siRNA-mediated knockdown led to reduced sensitivity to all of these inhibitors (Supplementary Fig. 8a, b). Consistent with our earlier data suggesting that AKT and SGK1 activation was not required for the effects of p85β, p85β-overexpressing cells were not more sensitive to an AKT (MK-2206) or SGK1/2 inhibitor (GSK650394) (Fig. 7b, c). We also investigated the in vivo anti-tumor effect of AXL inhibition (TP-0903) on p85β-overexpressing or vector control-expressing DOV13 ovarian cancer xenografts (Supplementary Fig. 9a), which reproduced serous histology consistent with its origin and previous report[37,38]. The p85β-overexpressing tumors showed significant growth inhibition after treatment with TP-0903 compared with the control tumors, which were not responsive to TP-0903 (*P* < 0.05; Fig. 7d). Protein was extracted from the tumors and western blotting demonstrated decreased AXL phosphorylation in all TP-0903-treated mice (Supplementary Fig. 9b). The tumors developed from p85β-overexpressing cells preserved the signaling features observed in vitro, including stronger phosphorylation of PDK1 and SGK3 but not AKT or mTOR than that in the vector control-expressing tumors (Supplementary Fig. 9b). Moreover, TP-0903 effectively decreased p-PDK1 and p-SGK3. Together, these findings suggest that therapeutic blockade of AXL signaling may be an effective strategy for ovarian tumors with *PIK3R2* amplification.

## Discussion
AXL is overexpressed in multiple cancer types, including ovarian cancer, causing malignant phenotypes and a poor prognosis[39–41]. Genomic aberrations in AXL are relatively rare in cancers. It

appears that AXL is primarily regulated at the levels of the promoter[42–44] and protein stability[12,45–47]. In this study, we showed that p85β but not p85α regulated the autophagy-lysosomal machinery to promote AXL protein stabilization and downstream signaling (Fig. 7e). Previous studies have suggested that the E3 ligase chromatin immunoprecipitation promotes the ubiquitination-mediated proteasomal degradation of AXL[47], whereas cbl-b targets all three members of TAM (Tyro3, AXL, and MERTK) for ubiquitination[46]. Monoubiquitination of AXL has been reported, but the ligase involved has yet to be identified[12]. We demonstrated that AXL was ubiquitinated by TRIM2, which had not been demonstrated to target any RTK. Importantly, AXL was a selective TRIM2 target among TAM family members, contributing to selective regulation of AXL by p85β. A second layer of regulation of AXL lies in the phosphorylation of optineurin. Several optineurin phosphorylation sites have been previously characterized. Data so far have indicated that these sites may relate to distinct functions. For example, S177 facilitates autophagosome formation at least partly by enhancing LC3 binding[23,26], whereas S473 promotes the binding of optineurin to ubiquitin[24,25]. Unphosphorylated optineurin preferentially binds polyubiquitin chains, not monoubiquitin[25]. Intriguingly, phosphorylated optineurin at S473 displays enhanced binding to monoubiquitin with an affinity comparable to that for polyubiquitin chains[25], therefore widening the spectrum of ubiquitinated targets that can be recognized by optineurin. Our data demonstrated that mutation of S473 caused an effect on AXL level similar to that of mutating S526. Further, combined mutation of the two residues was not additive. It remains to be examined whether S526 also alters the binding capacity of optineurin to ubiquitin. Moreover, whether p85β regulates optineurin phosphorylation at S473 is not known, because phosphorylation of this residue could not be detected in our samples. Accordingly, a low endogenous level of S473 phosphorylation has been previously reported[24,27]. The other optineurin phosphorylation site we investigated is S513, which did not alter the effect of p85β on AXL. Studies have shown that S513 alone does not appear to regulate ubiquitin binding[24,27] but that S513 together with S473 may mediate mitophagy by promoting the recruitment and retention of optineurin on damaged mitochondria[24,27].

p85 was not previously linked to regulation of the autophagy-lysosomal pathway. How p85β regulates the phosphorylation of TRIM2 and optineurin remains to be elucidated. The effect of p85β on AXL is likely mediated by the interaction of p85β with a set of proteins that do not interact coordinately with p85α, which may indeed also account for the differential activities of p85α and p85β. Nevertheless, p110 activity and SGK3/PDK1 signaling are not involved, because inhibition of these molecules had no effect on AXL regulation. The previously identified kinases of optineurin include TBK1, IKKβ, and Plk1[27,48,49]. The potential roles of these kinases in mediating the regulatory effect of p85β on AXL warrant

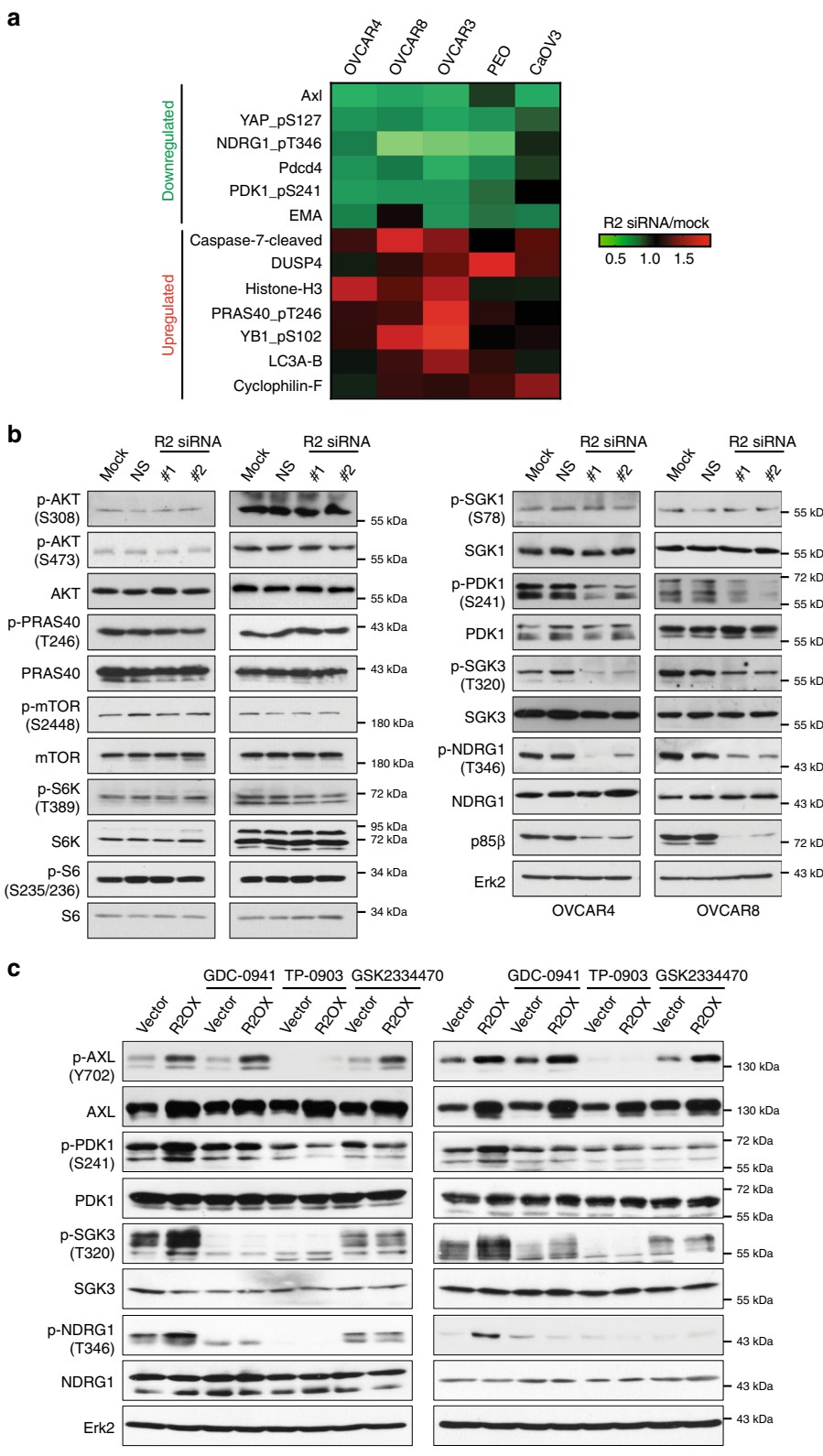

**Fig. 6 p85β activates the AXL/PDK1/SGK3 signaling axis. a** Heatmap illustrating proteins downregulated (<0.8) or upregulated (>1.2) in at least three ovarian cancer cell lines transfected with *PIK3R2* siRNA, compared with the mock control by RPPA analysis. **b** Lysates from cells transfected with *PIK3R2* siRNA for 72 h were immunoblotted using the indicated antibodies. NS, nonspecific siRNA. **c** Stable *PIK3R2*-overexpressing (R2OX) or vector control-expressing cells were treated with pan-p110 inhibitor GDC-0941 (10 μM), AXL inhibitor TP-0903 (0.5 μM), or PDK1 inhibitor GSK2334470 (2 μM) for 48 h. Lysates were subjected to western blotting. The western blotting experiments were repeated three times with independent lysates and results were reproducible. Source data are provided as a Source Data file.

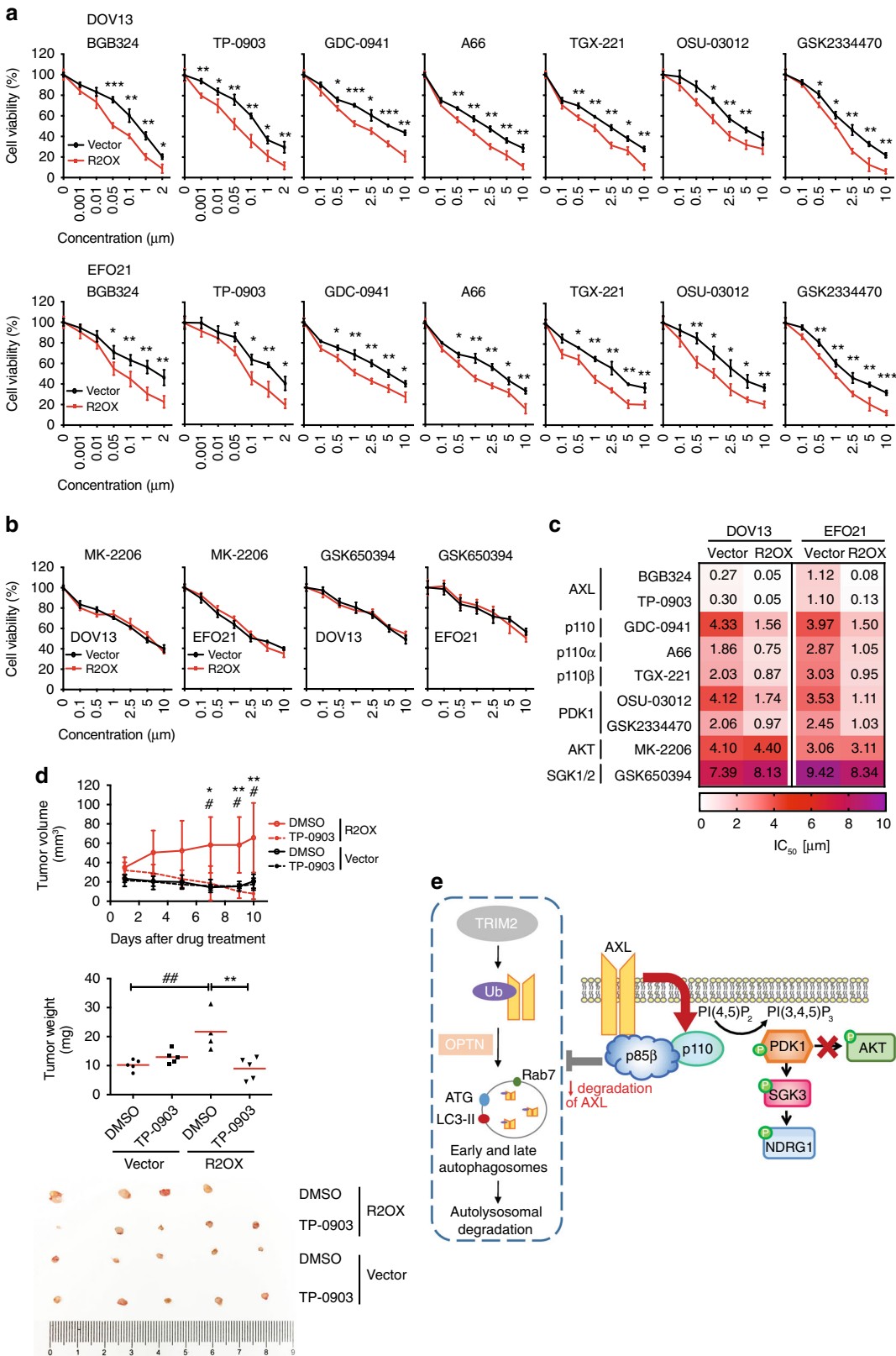

further investigation. Our data showed that the induction of AXL protein stability and phosphorylation by p85β was sustained even in the presence of Gas6. Further, the phosphorylation of AXL induced by p85β and Gas6 appeared to be additive, suggesting p85β and Gas6 as two independent activators of AXL. The level of Gas6 was shown to be elevated in ovarian cancer patients[50,51]. It

would be interesting to determine the co-occurrence of Gas6 elevation and *PIK3R2* amplification, as well as the corresponding AXL activation status in cancer patients. In ovarian cancer, an increased *PIK3CA* copy number is common (79%; $n = 579$), but *PIK3CA* mutation is rare (2 cases out of 316)[52]. Remarkably, *PIK3R2* amplification has been detected in 81% of non-*PIK3CA*-

**Fig. 7 Cells with p85β expression are sensitive to inhibitors of AXL/PDK1 signaling. a, b** DOV13 and EFO21 cells with or without stable *PIK3R2* overexpression were cultured for 7 days to allow 3D spheroid formation prior to treatment with the indicated inhibitors for another 72 h. Dose–response curves of each inhibitor are shown. **a** Inhibitors targeting AXL, pan-p110, p110α, p110β, and PDK1; **b** Inhibitors targeting AKT and SGK1/2. Data shown represent mean ± SD (n = 3 biologically independent samples). *P < 0.05, **P < 0.01, ***P < 0.001 by two-tailed t-test. **c** Heatmap illustrating IC50 values of the inhibitors in the treated cells. **d** DOV13 cells with or without stable *PIK3R2* overexpression were injected subcutaneously into female nude mice. Tumors were allowed to establish for 6 weeks before the mice were randomized into two subgroups for treatment with either DMSO or AXL inhibitor TP-0903 (50 mg/kg) for 10 days. Tumor volumes were measured over time (upper). Tumor weight (middle) and images of the dissected tumors (lower) after collection are shown. Data are shown as the mean ± SD (n = 5). *P < 0.05; **P < 0.01 for the comparison between DMSO- and TP-0903-treated R2OX subgroups, whereas #P < 0.05; ##P < 0.01 for the comparison between R2OX and Vector with DMSO treatment by two-tailed t-test. Source data are provided as a Source Data file. **e** Proposed model of the mechanism underlying the oncogenicity of p85β: p85β inhibits AXL degradation through autolysosome pathway, in which TRIM2 is an E3 ligase mediating AXL ubiquitination, while optineurin activates autophagy mediating degradation of the ubiquitinated AXL. Increase in AXL protein level induced by p85β activates PI3K-PDK1-SGK3-NDRG1 signaling instead of AKT.

amplified tumors compared with 52% of *PIK3CA*-amplified tumors (*P* < 0.0001 by McNemar's test). However, whether *PIK3CA* amplification leads to the same downstream effects as *PIK3R2* amplification remains to be fully characterized.

In summary, we describe herein that p85β upregulates the level of the upstream kinase AXL. The rationale for AXL inhibition in cancer therapy so far has been the frequent overexpression of AXL and the associated poor prognosis in cancers. Our findings that AXL mediates the oncogenic signaling of p85β provide further support for targeting AXL as a cancer treatment and suggest that *PIK3R2* amplification could serve as a biomarker for the efficacy of AXL inhibitors.

## Methods

**Cell lines and inhibitors**. OVCAR8, OVCAR4, OVCAR3, EFO21, and DOV13 were obtained from National Cancer Institute. SKOV3 was from American Type Culture Collection. The cell lines were maintained in RPMI-1640 media (Gibco, Carlsbad, CA) supplemented with 5% fetal bovine serum (FBS) (Gibco), and 1% penicillin–streptomycin at 37 °C and 5% $CO_2$. Cell lines were authenticated using short tandem repeat DNA profiling and tested with negative mycoplasma contamination. PI3K inhibitors (GDC-0941, PIK-90, A66, BYL719, and TGX-221), AKT inhibitors (MK-2206 and GDC-0068), and AXL inhibitor (BGB324) were obtained from Selleckchem (Houston, TX). AXL inhibitor (TP-0903) was purchased from ApexBio (Houston, TX). PDK1 inhibitors (OSU-03012 and GSK2334470) and SGK1/2 inhibitor (GSK650394) were purchased from Santa Cruz Biotechnology (Dallas, TX). The used doses of the inhibitors in vitro are as follows: GDC-0941, 10 μM; PIK-90, 10 μM; A66, 2 μM; BYL719, 2 μM; TGX-221, 10 μM; MK-2206, 5 μM; GDC-0068, 5 μM; BGB324, 1 μM; TP-0903, 0.5 μM; OSU-032012, 5 μM; GSK2334470, 2 μM; and GSK650394, 10 μM.

**siRNA, shRNA, and plasmids**. ON-TARGETplus siRNA targeting human *PIK3R2*, SMARTpool siRNA targeting the 14 proteins in degradative pathways, and nonspecific siRNA were obtained from Dharmacon (Lafayette, CO). Additional individual siRNA against *TRIM2*, *OPTN*, and *NEDD4* was purchased from Integrated DNA Technologies (Coraville, IA). siRNA transfection was performed using Lipofectamine RNAiMAX (Life Technologies, Carlsbad, CA). OVCAR8 and SKOV3 with stable *PIK3R2* knockdown and the corresponding vector control were established by infection of lentivirus with pLKO.1-*PIK3R2* shRNA or pLKO.1 vector prior to puromycin selection. DOV13 and EFO21 cells with stable *PIK3R2* overexpression or vector control were constructed by infection of lentivirus with pLenti6-*PIK3R2* or empty pLenti6 vector followed by blasticidin selection. AXL and DN-AXL plasmids were kind gifts from Professor Axel Ullrich (Max Planck Institute of Biochemistry, Germany). Expression plasmids of human *TRIM2* in pHBLV lentiviral vector was purchased from Genemedi (Shanghai, China) and human *OPTN* in pEGFP-N1 was from Addgene (Watertown, MA). Mutants of *OPTN* or *TRIM2* were generated using QuikChange Lightning Site-Directed Mutagenesis kit (Agilent Technologies, Santa Clara, CA). Plasmids were introduced to cells using Lipofectamine™ 3000 reagent according to the manufacturer's protocol. All constructs were validated by DNA sequencing. Sequence information of siRNA and shRNA was shown in Supplementary Table 2.

**Proliferation assay**. Cells, with or without treatment, were seeded at the density of 1000 cells/well in triplicate in 96-well plates for 72 h. Cell proliferations were detected using BrdU (bromodeoxyuridine) cell proliferation assay kit (Cell Signaling) according to the manufacturer's instruction. Experiments were performed three times independently.

**Colony formation assay**. DOV13 and EFO21 *PIK3R2*-stably expressing cells were either treated with inhibitors or transfected with DN-AXL/vector and were seeded into 6-well plates at a density of 200 cells/well in triplicate. Likewise, OVCAR4, OVCAR8, and SKOV3 cells were transfected with *PIK3R2* siRNA or nonspecific siRNA for 24 h prior to cell seeding. After 14 days, colonies were washed twice with phosphate-buffered saline (PBS), fixed with methanol for 10 min, and stained with crystal violet for 15 min at room temperature. Experiments were performed three times independently.

**Invasion assay**. Stable cells or siRNA-transfected cells were suspended in serum-free medium and seeded into insert with pore size of 8.0 μm (Millipore, Billerica, MA) precoated with 1 mg/mL Matrigel (Corning, NY). Cell culture medium containing 10% FBS were added to the lower chambers and cells were incubated for 24 h. Invasive cells were fixed by methanol and stained with crystal violet. The number of invasive cells was counted in five randomly captured fields using light microscope (×100). All assays were performed in triplicate.

**Cell viability assay**. Cells were seeded at a density of 1000 cells/well in triplicate in 96-well plates. Culture medium was replaced at day 3. Afterwards, cells were incubated with 10 μL Resazurin solution (0.2 mg/mL; Cell Signaling, Danvers, MA) for 3 h prior to measurement of absorbance 600 nm. Experiments were performed three times independently.

**In vivo tumorigenicity assay**. The protocols were approved by the Committee on the Use of Live Animals in Teaching and Research of The University of Hong Kong and all mouse experiments were conducted according to ethical regulations. OVCAR8 or SKOV3 cells ($5 \times 10^6$) with stable *PIK3R2* knockdown or vector control in sterile PBS were injected intraperitoneally into 6-week-old female athymic BALB/c mice (n = 5 per group) (Charles River Lab, USA). Six weeks after injection, mice were killed and the disseminated tumor nodules in the peritoneal cavity were quantified and weighed.

To evaluate the antitumor effect of TP-0903 (AXL inhibitor), $1.5 \times 10^7$ DOV13 cells stably expressing *PIK3R2* or vector control were mixed with Matrigel (2:1) and implanted subcutaneously into the right flank of each mouse. Four weeks after inoculation, mice were randomly divided into two subgroups and orally administrated with 50 mg/kg TP-0903 (n = 5) or vehicle (5% dimethylsulfoxide (DMSO), n = 5) daily for 10 days. Tumor volume was calculated as length × width × width × 0.5 (length is the major tumor axis, whereas width is the minor axis). One mouse injected with *PIK3R2*-expressing cells treated with DMSO died during the treatment period and was excluded from the analysis. Tumor burden based on tumor volumes and weight was compared among subgroups. Protein collected from the tumor samples was used for western blotting. Some tumor nodules were fixed and were subjected to hemotoxylin and eosin staining for histopathology.

**3D spheroid-based drug sensitivity assay**. *PIK3R2* siRNA-transfected or *PIK3R2*-overexpressing cells were suspended in 2% Matrigel-containing medium and seeded at a density of 1000 cells/well in triplicate in Matrigel-precoated 96-well plates. Cells were cultured for 7 days for 3D spheroids formation, followed by treatment of inhibitors at serial concentrations for 72 h. Luminescent readout as viability indicator was detected by CellTiter-Glo® 3D cell viability assay according to the manufacturer's instruction (Promega, Madison, WI). The relative percentage of cell viability after inhibitor treatment was shown in the dose–response curves with the untreated cells as 100%. IC50 values of tested drugs were calculated by nonlinear regression analysis using GraphPad Prism 7 (La Jolla, CA). Assays were performed in triplicate.

**Real-time PCR**. Five micrograms of total RNA isolated using TRIzol reagent (Life Technologies) were reverse transcribed using SuperScript IV and oligo (dT) primers (Life Technologies), followed by cDNA amplification using Power SYBR

Green PCR Master Mix system according to the manufacturer's instructions (Applied Biosystems, Foster City, CA). *GAPDH* was used as an internal control to calculate the relative *AXL* mRNA levels. Primer sequences are listed in Supplementary Table 3. All assays were performed in triplicate.

**Luciferase assay**. pGL3 luciferase plasmid with human *AXL* gene promoter (2376 bp) was kindly provided by Professor Heike Allgayer at the Mannheim Medical Faculty of University of Heidelberg, Germany. Cells transfected with siRNA for 24 h were transfected again with pGL3 basic reporter vector or pGL3-AXL with *Renilla* luciferase plasmid (pRL-TK). After 48 h, firefly and *Renilla* luciferase activities (for normalization) were measured by Dual-Luciferase Reporter Assay System (Promega). Assays were performed in triplicate.

**Protein extraction, immunoprecipitation, and immunoblotting**. Total protein for western blotting was prepared using RIPA lysis buffer (25 mM Tris-HCl pH 7.4, 150 mM NaCl, 0.1% SDS, 1% NP-40, 1% sodium deoxycholate) supplemented with protease and phosphatase inhibitors (Thermo Fisher Scientific, Waltham, MA). After SDS-polyacrylamide gel electrophoresis, membranes were probed with indicated primary antibodies overnight at 4 °C, horseradish peroxidase (HRP)-conjugated secondary antibodies for 1 h at room temperature and detected with Western ECL kit (Bio-Rad, Hercules, CA). ERK2 was used for normalization. For immunoprecipitation, cells were lysed in buffer containing 50 mM Tris, 150 mM NaCl, 0.5% NP-40, 5 mM EDTA, protease, and phosphatase inhibitors. The lysates were incubated with antibody overnight followed by incubation with Protein A/G (Santa Cruz Biotechnology) for 4 h. Following beads washing and elution by 2× Laemmli sample buffer (62.5 mM Tris-HCl pH 6.8, 25% glycerol, 2% SDS, 0.01% bromophenol blue, 5% β-mercaptoethanol), the immunocomplexes were analyzed by western blotting. The information of all antibodies used is shown in Supplementary Table 4.

**Immunofluorescent staining**. OVCAR8 cells ($3 \times 10^4$) seeded onto sterilized cover slips were transfected with *PIK3R2* siRNA for 48 h. Cells were then fixed with 4% paraformaldehyde and permeabilized with 0.1% Triton X-100. After blocking with 3% bovine serum albumin, cells were incubated with primary antibody overnight (Supplementary Table 4). After washing with TBST buffer, cells were incubated with secondary antibody for 1 h. Images were captured with Carl Zeiss LSM 700 (Zeiss, Jena, Germany).

**Proximity ligation assay**. Cells seeded onto sterilized cover slips were subjected to Duolink PLA fluorescence according to the manufacturer's instruction (Sigma-Aldrich, St. Louis, MO). Prior to incubation with two primary antibodies overnight (Supplementary Table 4), cells were fixed by 4% paraformaldehyde and permeabilized by 0.1% Triton X-100. Negative control was performed by replacing one of the antibodies with IgG. The cells were then incubated with PLA probe (PLUS and MINUS) for 1 h. Probe ligation were performed, followed by polymerase-mediated amplification of PLA signals. Fluorescence signals were observed at 461 nm (4′,6-diamidino-2-phenylindole) and 594 nm (Texas Red) by Carl Zeiss LSM 700 (Zeiss). At least eight fields of each slide were captured and the number of spots per nucleus was counted. Experiments were performed three times independently.

**PI3K activity assay**. PI3K catalytic subunit p110α or p110β was isolated by immunoprecipitation and their kinase activities were assessed by measurement of PI(3,4,5)P3 generated from PI(4,5)P2 substrate solution using the competitive in vitro PI3-Kinase Activity ELISA Pico (Echelon Biosciences, Salt Lake City, UT) according to the manufacturer's indication. PIP3 production was calculated by the standard curve using sigmoidal dose–response (variable slope) correlation and PI3-kinase activities were indicated by relative PIP3 levels normalized with corresponding control. All assays were performed in triplicate.

**PI(3,4,5)P3/PI(4,5)2 quantification**. After treatment, cells were collected in ice-cold 0.5 M trichloroacetic acid. The organic phases containing acidic lipids were collected as reported previously[4] and reconstituted by PBST for PIP3 and PIP2 detection using ELISA kits (K2500 and K4500, Echelon Biosciences). Relative PIP3 levels of samples were normalized by PIP2 levels in parallel. All experiments were performed in triplicate.

**Phosphoproteome sample preparation and MS analyses**. Sample processing and liquid chromatography tandem MS (LC-MS/MS) were performed at LKS Faculty of Medicine, Proteomics, and Metabolomics Core Facility, Faculty of Medicine, University of Hong Kong. A total of 1.5 mg of protein lysate per sample was digested by the FASP method[53]. For total proteome analysis, 50 μg of the peptides was used and the remaining were subjected to phosphopeptide enrichment using TiO$_2$ beads[54]. The peptides were desalted using C18 ziptips. Total and enriched peptides were subjected to LC-MS/MS analyses on Orbitrap Fusion Lumos ETD mass spectrometer interfaced with Dionex 3000RSLC nanoLC. The high-resolution, high-mass accuracy MS data obtained were processed using Maxquant version 1.6.0.1. MS data analyzed in triplicate for each condition were searched using inbuilt Andromeda algorithm against Uniprot human protein database (95,000 entries). Search parameters–Enzyme specificity was set to trypsin, up to two missed cleavages allowed for digestion, cysteine carbamidomethylation as a fixed modification, and phosphorylation of Ser, Thr, and Tyr, and oxidation of methionine as variable modifications. Phosphorylated site quantification was calculated using the label free quantification intensities. Data visualization and bioinformatic analysis was performed in the Perseus software version 1.6.2.1. For statistical analysis, replicates were grouped and analysis of variance test was carried out with permutation-based 1% false discovery rate. Hierarchical clustering of phosphosites was performed on logarithmized intensities.

**Reverse-phase protein array**. RPPA was carried out by our group[55,56]. Cells were lysed in lysis buffer containing 1% Triton X-100, 50 mM HEPES (pH 7.4), 150 mM NaCl, 1.5 mM MgCl$_2$, and 1 mM EGTA. Protein concentration was adjusted to 1.5 μg/μl and denatured protein in 4× SDS buffer was printed onto nitrocellulose-coated slides. The slides were incubated with primary antibodies. Signal was visualized by HRP and DAB colorimetric reaction. Slides were scanned for the quantification of signal intensities, which were determined by "Supercurve Fitting" with a logistic regression model. All data points were normalized for protein loading.

**Immunohistochemistry**. Human ovarian tumor tissue array (OVC1021) was obtained from Pantomics (Richmond, CA). The slides were deparaffinized and rehydrated in graded ethanol prior to antigen retrieval using citrate buffer pH 6.0. They were then incubated with 3% H$_2$O$_2$ to quench endogenous peroxidase activity and goat serum for blocking. Incubation with primary antibody (1:25 for p85β and 1:40 for AXL) was performed at 4 °C overnight and subsequently with biotin-conjugated secondary antibody (Dako, Carpinteria, CA) for 1 h at room temperature. HRP was detected by DAB (Amresco, Solon, OH). Protein levels were represented by histoscore scores on an arbitrary scale: 0, no immunoreactivity; 1, weak; 2, moderate; 3, intense; and 4, very intense.

**Statistical analyses**. All experiments were repeated three times. Experiment data were presented as mean ± SD and *P*-values were calculated using Student's unpaired two-tailed *t*-test (unless indicated otherwise) by Prism software. *P*-values < 0.05 were considered significant.

**Reporting summary**. Further information on research design is available in the Nature Research Reporting Summary linked to this article.

## Data availability

Source Data file contains the raw data underlying all reported averages in graphs and uncropped versions of blots presented in the figures. The mass spectrometry proteomics data have been deposited to the ProteomeXchange Consortium via the PRIDE[57] partner repository with the dataset identifier PXD018449. Data pertaining to TCGA ovarian cancer samples were obtained from the cBioPortal (https://www.cbioportal.org). Additional information is available from the authors upon reasonable request.

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

## Acknowledgements

This study was supported by Hong Kong Research Grants Council (17111618) to L.W.C. We thank the RPPA Core Facility (funded by NCI #CA016672) of the MD Anderson Cancer Center (Houston, TX) and the Faculty Core Facility of the LKS Faculty of Medicine HKU for the help with confocal microscopy. G.B.M. is supported by a kind gift from the Adelson Medical Research Foundation, the Ovarian Cancer Research Foundation, The Breast Cancer Research Foundation, The Komen Foundation SAC110052, and NCI grants CA217685, CA217842, and CA098258. We also thank Professor Axel Ullrich (Max Planck Institute of Biochemistry, Germany) for generously providing the AXL plasmids and Professor Heike Allgayer (University of Heidelberg, Germany) for the human AXL gene promoter.

## Author contributions

L.W.C. conceived and coordinated the project. L.R., V.C.M., and L.W.C. designed the experiments. L.R., V.C.M., Y.Z., D.Z., X.L., C.C.F., C.G., Y.L., and L.W.C. performed experiments. R.S. contributed to mass spectrometry analysis. L.R., V.C.M., G.L.T., A.N.C., G.B.M., and L.W.C analyzed the data. L.W.C. wrote the manuscript, with input from the other authors.

## Competing interests

G.B.M. consults with AstraZeneca, ImmunoMET, Ionis, Nuevolution, PDX bio, Signalchem, Symphogen, and Tarveda; has stock options with Catena Pharmaceuticals, ImmunoMet, SignalChem, Spindle Top Ventures, and Tarveda; sponsored research from AstraZeneca, Immunomet, Pfizer, Nanostring, and Tesaro; travel support from Chrysallis Bio; and has licensed technology to Nanostring and Myriad Genetics. The other authors declare no competing interests.
