## [Peer Review File · Nature Communications]

Editorial Note: Parts of this Peer Review File have been redacted as indicated to remove confidential information.

Reviewers' comments:

Reviewer #1 (Remarks to the Author):

Ling Rao et al demonstrated in a very elegant way the role of PIK3R2 (encodes to p85beta) as an oncogene. Specifically, p85beta signal induces PDK1/SGK3 pathway through upstream kinase activity of AXL. Mechanistically they also showed that p85beta regulates AXL protein degradation.

The data is convincing, the MS is well written and simple to understand. I have only a few minor comments:

1. The statement of oncogenicity of p85 β is independent of AKT comes from the lack of anti-tumor effect of small molecules inhibitors on cells proliferation and colony formation. This data should be supported with signalling of the AKT/and its mTOR pathway, or/and KD of AKTs.
2. The data in figure 4 lacks key information of AXL level, and the on target effect of the drugs.
3. It will be useful to provide histopathology on the tumors, including for signalling of the tumor cells.

Reviewer #2 (Remarks to the Author):

In this work, Dr. Cheung and the colleagues aimed to mechanistically understand the oncogenic roles of PIK3R2/p85 β through molecularly dissecting its potential down-stream signaling pathway in the context of p85 β -p110 complex. The authors first discovered that PIK3R2/p85 β up-regulated the static level of AXL, while activating p110 but having no effect on the protein level of p110. They then went on to demonstrate that (supposedly phosphorylated) TRIM2 could serve as a specific E3 Ub ligase for AXL and OPTN acted as an autophagy receptor for ubiquitinated AXL, by showing that knockdown of either TRIM2 or OPTN, or both, could abolish the effect of p85 β on stability of AXL. Through showing that "increased AXL level by p85 β was abrogated upon optineurin overexpression (Fig. 3g)" but not the phosphorylation-defective optineurin mutant S526A, they went on to propose that phosphorylation of OPTN could promote its autophagy receptor function to facilitate the degradation of AXL. Finally, the authors demonstrated that p85 β might initiate oncogenic signaling through activating PDK1/SGK3/NDRG1 but not AKT. Overall, this study came to the current model with huge amounts of efforts and multi-line of evidences to support the conclusion, with most of the experiments well designed and properly carried out.

However, there were still many holes that needed to be filled in when it came to the strength of the conclusions and the way to prove the points. Specifically, the following questions needed to be addressed, with importance not necessarily in numeric order:

- 1) When speaking of the phenomenon in which p85 β up-regulated the static level of AXL, it was often mentioned in the manuscript that "the expression of AXL" was up-regulated. Taking from the observation that chloroquine treatment reversed the decrease in AXL level induced by p85 β depletion (Fig. 3a-3b; Supplementary Fig. 4c-4d), ectopic expression of p85 β led to increase in the homeostatic level of AXL most likely through inhibiting the autophagic degradation of AXL, rather than augmenting the transcription of PIK3R2 or its mRNA translation. Therefore, it is really important to reword/correct the related expressions accordingly.
- 2) Data from the cropped image presented in Fig. 3c did not support the claim that AXL was mono-ubiquitinated by TRIM2. First, the ubiquitination should be repeated properly to determine whether it was indeed mono-Ub that TRIM2 conjugated onto AXL; secondly, it has been shown clearly and many

times in the literature that OPTN, phosphorylated or not, would preferentially bind to poly-Ub chain, often in linear (Met1) linkage. How could mono-ubiquitinated AXL be recognized by OPTN?

3) OPTN as well as many other autophagy receptors have been shown to be substrates for many kinases, e.g. TBK1 and ULK1 etc., and there were many phosphorylation sites in OPTN, and even in UBA1 domain of OPTN, phosphorylation on S473 rather than S526 was found to be more important in regulating its binding to Ub chains (Ref 21). It was thus totally unclear why the other single or joint phosphorylation mutants were not tested.

4) The last sections of the manuscripts were dedicated to the study of p85 β -mediated oncogenic cell signaling, in which the cascade of PDK1/SGK3/NDRG1 but not AKT was shown to be activated. It remained unclear whether and how the kinases activated in this cascade would be related to those still missing kinases that would phosphorylate TRIM2 and/or OPTN.

5) Language may need to be further polished for formal publication.

Therefore, I would recommend major revision before its final acceptance for publication.

Reviewer #3 (Remarks to the Author):

In this paper the authors analyse the impact of p85 β /PI3KR2 on the regulation of AXL protein levels. First, the authors analyse the genetic amplification and mRNA expression of PI3KR2 in ovarian cancer patients and observe a negative correlation with OS, which indicates that high levels of PI3KR2 confer a bad prognosis. So they deplete PI3KR2 by siRNA or shRNA in 3 ovarian cancer cell lines with high PI3KR2 levels and find a decrease in proliferation in vitro and in vivo in mice, and a reduction in invasion and colony formation. Conversely, if they overexpress PI3KR2 in cells with low endogenous expression they find an increase in proliferation, invasion and colony formation, which are reversed by inhibitors of PI3K. Overexpression or depletion of PI3KR2 correlate with the kinase activity of p110 (but not the total protein levels of p110) and the formation of PIP3.

They then find a correlation between the levels of PI3KR2 and AXL by RPPA analysis of TCGA samples and also by IHC. They verify by western blot that the levels of AXL and pAXL depend on the levels of PI3KR2 in cells that are depleted of or overexpress PI3KR2, and this is independent of the activity of p110. Inhibition of AXL with a DN mutant or an inhibitor completely block the invasion/proliferation/colony formation/p110 activation phenotypes promoted by PI3KR2 overexpression. This indicates that the effect of PI3KR2 is mediated by AXL.

Interestingly, the authors find that PI3KR2 expression affects the protein but not the mRNA levels of AXL and, if they inhibit lysosomal degradation, they observe no degradation of AXL in cells depleted of PI3KR2. Axl is ubiquitinated in cells where PI3KR2 is downregulated. By phospho proteomics analysis the authors find that autophagy is enriched and they focus their attention on Trim2 and Optn, whose depletion reverse AXL degradation in PI3KR2-depleted cells. Conversely, overexpression of active forms of Trim2 and Optn strongly reduces the high levels of AXL induced by the overexpression of PI3KR2. The depletion of PI3KR2 induces AXL interaction with Trim and Optn by IP. By immunofluorescence the authors observe that depletion of PI3KR2 induces colocalization of AXL with LC3B.

By RPPA analysis, the authors analyse the signalling pathways downstream of PI3KR2 and find that in cells depleted of PI3KR2 also pPDK1, pSGK3 and pNDRG1 levels are lower, while they increase with PI3KR2 overexpression. Inhibiting AXL or p110 with inhibitors block this effect.

In the last figure, data show that cells overexpressing PI3KR2 (and therefore with higher protein levels of AXL) are more sensitive to an AXL inhibitor in 3D spheroids and in a mouse model.

The data shown by the authors are solid and the correlation between PI3KR2 protein levels and AXL protein levels is well demonstrated with a multitude of assays in several cell lines. AXL is clearly degraded in the absence of PI3KR2 or stabilised when PI3KR2 is overexpressed and Trim2 and Optn seem to have a role in this. However, the immunofluorescence images are less convincing: while a colocalisation can be seen between AXL and LC3B (though this seems to happen only in one cell), the colocalisation with Rab7 is scarce and even less with Lamp1 and Lamp2, where it appears to be due to

a saturation of the fluorescent signal more than to the presence of both proteins on the same puncta. Can the author analyse colocalisation with other autophagic markers?

The data showing the inhibiting AXL or p110 with inhibitors reverse the effect of PI3KR2 overexpression on pPDK1, pSGK3 and pNDRG1 levels are not convincing and very variable, they should be either improved or removed if not solid enough.

My main concern is the model. The data from the authors strongly suggest that AXL is downstream of PI3KR2 and not upstream as shown in the final model. All the experiments have been performed in serum, without any AXL-specific stimulation and the phosphorylation of AXL observed by the authors could be also due to cross-phosphorylation by other RTKs. What happens if AXL is stimulated by GAS6? What is the role of PI3KR2 in this context? This is a big limitation and should be extensively addressed.

Also, no protein-protein interactions have been shown in this paper. Does AXL interact with PI3KR2, Trim2 and Optn? This could be tested by proximity ligation assay and co-IP. Does AXL activation by GAS6 affect these interactions? I believe that a deeper mechanistic insight could strengthen this paper and make it suitable for publication.

Minor comments:

- please indicate in the text where the western blots showing the depletion of PI3KR2 by siRNA or shRNA are located as they are not in figure 1 and suppl figure 1.

-please show the graphs of the correlation between PI3KR2 and AXL as only the stats are mentioned in the text.

Point-by-point responses to referees' comments

We sincerely thank the reviewers for their positive comments and constructive criticisms that have helped us improve the manuscript. We have performed substantial amount of new experiments to address the raised concerns. The responses are listed point-by-point below. Page, line and figure refer to the revised manuscript. We have marked the changes in the revised manuscript in red.

Reviewer #1 (Remarks to the Author):

Ling Rao et al demonstrated in a very elegant way the role of *PIK3R2* (encodes to p85beta) as an oncogene. Specifically, p85beta signal induces PDK1/SGK3 pathway through upstream kinase activity of AXL. Mechanistically they also showed that p85beta regulates AXL protein degradation.

The data is convincing, the MS is well written and simple to understand. I have only a few minor comments:

1. The statement of oncogenicity of p85 β is independent of AKT comes from the lack of anti-tumor effect of small molecules inhibitors on cells proliferation and colony formation. This data should be supported with signalling of the AKT/and its mTOR pathway, or/and KD of AKTs.

Response: Thank you for all the constructive suggestions. We have now included additional data accordingly to further examine the reliance of p85 β on AKT signaling. Knockdown experiment of AKT1/2/3 by siRNA was performed in two p85 β -overexpressing ovarian cancer cell lines. The phenotypes (cell proliferation, colony formation and cell invasion) induced by p85 β were not affected by the siRNA (**Supplementary Fig. 2d; page 4, lines 24-26**).

Phosphorylated levels of AKT, the AKT substrate PRAS40 as well as mTOR, p70S6K and S6 were evaluated by Western blotting (**page 8, lines 34-37 to page 9, lines 1-3**). Phosphorylated levels of these proteins were not altered in *PIK3R2*-depleted cells (**Fig. 6b**) or p85 β -overexpressing cells (**Supplementary Fig. 7a**). Indeed, we consistently observed a modest decrease in p-AKT (S473) in p85 β -overexpressing cells (Fig. 3a and Supplementary Fig. 7a). These data together support that the oncogenicity of p85 β is unlikely dependent of AKT.

2. The data in figure 4 lacks key information of AXL level, and the on target effect of the drugs.

Response: We have now included Western blots data of AXL protein level (**Fig. 6c**), which was not changed by the inhibitors, suggesting that these molecules are not involved in the regulation of AXL (**page 9, lines 19-22**).

Fig. 6c and **Supplementary Fig. 7b** showed the on-target effects of the inhibitors AXL (TP-0903), PDK1 (GSK2334470) and p110 (GDC-0941). Phosphorylation of AXL

and PDK1 or the kinase activity of p110 were inhibited by the respective inhibitors (**page 9, lines 14-16**).

3. It will be useful to provide histopathology on the tumors, including for signalling of the tumor cells.

Response: Ovarian cancer of serous histology is the focus of this study. The TCGA tumor samples, ovarian tumor slides for IHC staining as well as cell lines are all of serous subtype. We have performed hematoxylin and eosin staining, which confirmed serous histopathology of the tumors developed in the xenografts (**Supplementary Fig. 9a; page 10, lines 6-7**).

Signaling of these tumors was also analyzed by Western blotting after extraction of proteins from the tumor nodules. The xenograft tumors essentially retained the signaling features of the cell lines in vitro. The activation of PDK1 and SGK3, but not AKT or mTOR, in p85 β -overexpressing tumors was evident compared to vector control-expressing tumors (**Supplementary Fig. 9b; page 10, lines 12-15**).

Reviewer #2 (Remarks to the Author):

In this work, Dr. Cheung and the colleagues aimed to mechanistically understand the oncogenic roles of PIK3R2/p85 β through molecularly dissecting its potential down-stream signaling pathway in the context of p85 β -p110 complex. The authors first discovered that PIK3R2/p85 β up-regulated the static level of AXL, while activating p110 but having no effect on the protein level of p110. They then went on to demonstrate that (supposedly phosphorylated) TRIM2 could serve as a specific E3 Ub ligase for AXL and OPTN acted as an autophagy receptor for ubiquitinated AXL, by showing that knockdown of either TRIM2 or OPTN, or both, could abolish the effect of p85 β on stability of AXL. Through showing that “increased AXL level by p85 β was abrogated upon optineurin overexpression (Fig. 3g)” but not the phosphorylation-defective optineurin mutant S526A, they went on to propose that phosphorylation of OPTN could promote its autophagy receptor function to facilitate the degradation of AXL.

Finally, the authors demonstrated that p85 β might initiate oncogenic signaling through activating PDK1/SGK3/NDRG1 but not AKT. Overall, this study came to the current model with huge amounts of efforts and multi-line of evidences to support the conclusion, with most of the experiments well designed and properly carried out.

However, there were still many holes that needed to be filled in when it came to the strength of the conclusions and the way to prove the points. Specifically, the following questions needed to be addressed, with importance not necessarily in numeric order:

1) When speaking of the phenomenon in which p85 β up-regulated the static level of AXL, it was often mentioned in the manuscript that “the expression of AXL” was up-regulated. Taking from the observation that chloroquine treatment reversed the decrease in AXL

level induced by p85 β depletion (Fig. 3a-3b; Supplementary Fig. 4c-4d), ectopic expression of p85 β led to increase in the homeostatic level of AXL most likely through inhibiting the autophagic degradation of AXL, rather than augmenting the transcription of *PIK3R2* or its mRNA translation. Therefore, it is really important to reword/correct the related expressions accordingly.

Response: Thank you for pointing this out. We have gone through the entire manuscript carefully and have revised the related expressions to “AXL protein level”.

2) Data from the cropped image presented in Fig. 3c did not support the claim that AXL was mono-ubiquitinated by TRIM2. First, the ubiquitination should be repeated properly to determine whether it was indeed mono-Ub that TRIM2 conjugated onto AXL; secondly, it has been shown clearly and many times in the literature that OPTN, phosphorylated or not, would preferentially bind to poly-Ub chain, often in linear (Met1) linkage. How could mono-ubiquitinated AXL be recognized by OPTN?

Response: We thank the Reviewer for these valuable comments. We have repeated the experiments three times using two ubiquitin-specific antibodies. They are clones FK2 and P4D1 which recognizes both the monoubiquitin and polyubiquitin moieties. Whole gel images are now presented in **Fig. 4c** (shorter blot exposure) and **Supplementary Fig. 5e** (longer blot exposure)(**page 6, lines 20-29**). We observed an increase in ubiquitinated AXL in the presence of *PIK3R2* siRNA. In addition to distinct bands which may correspond to mono-ubiquitinated AXL, we observed weak smear at the higher molecular weight which may represent multiple monoubiquitinated or polyubiquitinated AXL after prolonged blot exposure. We attempted to further examine the ubiquitination using the anti-ubiquitin antibody clone FK1 which is exclusively against polyubiquitin but not monoubiquitin.

[REDACTED]

Without a definitive call on the type of AXL ubiquitination, we have therefore removed our description of ubiquitinated AXL as mono-ubiquitination.

How optineurin recognizes monoubiquitinated AXL remains to be elucidated. The phosphorylation site (S526) identified in our mass spectrometry has not been identified and characterized before. However, it has been shown that phosphorylation of optineurin at S473 confers optineurin an ability to recognize and bind to both monoubiquitin and polyubiquitin (*Ref. 1*). Whether S526 phosphorylation also widens the ubiquitin-binding capacity of optineurin remains to be investigated. This has been added to the Discussion (**page 11, lines 18-25**).

3) OPTN as well as many other autophagy receptors have been shown to be substrates for many kinases, e.g. TBK1 and ULK1 etc., and there were many phosphorylation sites in OPTN, and even in UBAN domain of OPTN, phosphorylation on S473 rather than S526 was found to be more important in regulating its binding to Ub chains (Ref 21). It was thus totally unclear why the other single or joint phosphorylation mutants were not tested.

Response: We are grateful to the Reviewer for this suggestion. We have examined the effect of several other previously characterized phosphorylation sites of optineurin including S177, S473 and S513 (**page 8, lines 5-12**). Single mutation of S473, but not S177 or S513, could reverse the degradation of AXL (**Supplementary Fig. 6e**). However, combined mutation of S473 and S526 did not yield any additional effect. S473 could not be detected in our mass spectrometry analysis (probably because of low abundance) and therefore we do not know whether p85 β also regulates optineurin phosphorylation at S473. Accordingly, two previous independent studies have indicated that the level of S473 is low compared to some other phosphorylation sites (*Ref. 2, 3*). We have obtained a commercial antibody of optineurin S177 and our data showed that S177 level was unaltered by p85 β (**Supplementary Fig. 6f**). The data are now discussed (**page 11, lines 14-32**).

4) The last sections of the manuscripts were dedicated to the study of p85 β -mediated oncogenic cell signaling, in which the cascade of PDK1/SGK3/NDRG1 but not AKT was shown to be activated. It remained unclear whether and how the kinases activated in this cascade would be related to those still missing kinases that would phosphorylate TRIM2 and/or OPTN.

Response: The kinases that mediate the phosphorylation of TRIM2 and optineurin remain to be identified. The p110 and PDK1 signaling, however, are unlikely the mediators because inhibition of the signaling had no effect on AXL protein level (**Fig. 6c; page 9, lines 19-22**). We agree that it will be interesting in future work to identify these kinases, and we have added this to the Discussion (**page 11, lines 37-38 to page 12, lines 1-3**).

5) Language may need to be further polished for formal publication.

Response: We have got our revised manuscript further polished by professional language editing service.

References:

1. Li, F. *et al.* Structural insights into the ubiquitin recognition by OPTN (optineurin) and its regulation by TBK1-mediated phosphorylation. *Autophagy*. 14, 66-79 (2018).
2. Heo, JM. *et al.* The PINK1-PARKIN Mitochondrial Ubiquitylation Pathway Drives a Program of OPTN/NDP52 Recruitment and TBK1 Activation to Promote Mitophagy. *Mol. Cell*. 60, 7-20 (2015).
3. Richter, B. *et al.* Phosphorylation of OPTN by TBK1 enhances its binding to Ub chains and promotes selective autophagy of damaged mitochondria. *Proc. Natl. Acad. Sci. USA* 113, 4039-4044 (2016).

Reviewer #3 (Remarks to the Author):

In this paper the authors analyse the impact of p85 β /PI3KR2 on the regulation of AXL protein levels.

First, the authors analyse the genetic amplification and mRNA expression of PI3KR2 in ovarian cancer patients and observe a negative correlation with OS, which indicates that high levels of PI3KR2 confer a bad prognosis. So they deplete PI3KR2 by siRNA or shRNA in 3 ovarian cancer cell lines with high PI3KR2 levels and find a decrease in proliferation in vitro and in vivo in mice, and a reduction in invasion and colony formation. Conversely, if they overexpress PI3KR2 in cells with low endogenous expression they find an increase in proliferation, invasion and colony formation, which are reversed by inhibitors of PI3K. Overexpression or depletion of PI3KR2 correlate with the kinase activity of p110 (but not the total protein levels of p110) and the formation of PIP3.

They then find a correlation between the levels of PI3KR2 and AXL by RPPA analysis of TCGA samples and also by IHC. They verify by western blot that the levels of AXL and pAXL depend on the levels of PI3KR2 in cells that are depleted of or overexpress PI3KR2, and this is independent of the activity of p110. Inhibition of AXL with a DN mutant or an inhibitor completely block the invasion/proliferation/ colony formation/p110 activation phenotypes promoted by PI3KR2 overexpression. This indicates that the effect of PI3KR2 is mediated by AXL.

Interestingly, the authors find that PI3KR2 expression affects the protein but not the mRNA levels of AXL and, if they inhibit lysosomal degradation, they observe no degradation of AXL in cells depleted of PI3KR2. Axl is ubiquitinated in cells where PI3KR2 is downregulated. By phospho proteomics analysis the authors find that autophagy is enriched and they focus their attention on Trim2 and Optn, whose depletion reverse AXL degradation in PI3KR2-depleted cells. Conversely, overexpression of active forms of Trim2 and Optn strongly reduces the high levels of AXL induced by the overexpression of PI3KR2. The depletion of PI3KR2 induces AXL interaction with Trim and Optn by IP. By immunofluorescence the authors observe that depletion of PI3KR2 induces colocalization of AXL with LC3B. By RPPA analysis, the authors analyse the signalling pathways downstream of PI3KR2 and find that in cells depleted of PI3KR2 also pPDK1, pSGK3 and pNDRG1 levels are lower, while they increase with PI3KR2 overexpression. Inhibiting AXL or p110 with inhibitors block this effect. In the last figure, data show that cells overexpressing PI3KR2 (and therefore with higher protein levels of AXL) are more sensitive to an AXL inhibitor in 3D spheroids and in a mouse model.

The data shown by the authors are solid and the correlation between PI3KR2 protein levels and AXL protein levels is well demonstrated with a multitude of assays in several cell lines. AXL is clearly degraded in the absence of PI3KR2 or stabilised when PI3KR2 is overexpressed and Trim2 and Optn seem to have a role in this. However, the immunofluorescence images are less convincing: while a colocalisation can be seen between AXL and LC3B (though this seems to happen only in one cell), the colocalisation with Rab7 is scarce and even less with Lamp1 and Lamp2, where it appears to be due to a saturation of the fluorescent signal more than to the presence of both proteins on the same puncta. Can the author analyse colocalisation with other autophagic markers?

Response: We acknowledge the Reviewer's suggestion of other autophagic markers for colocalization with AXL. We have replaced the colocalization data involving LAMP1/2 with the autophagic markers ATG5 and ATG12 (**Fig. 5h; page 8, lines 19-20**). *PIK3R2* depletion promoted the colocalization of AXL with ATG5 or ATG12.

The data showing the inhibiting AXL or p110 with inhibitors reverse the effect of *PI3KR2* overexpression on pPDK1, pSGK3 and pNDRG1 levels are not convincing and very variable, they should be either improved or removed if not solid enough.

Response: We have repeated the experiments again and the old data have been replaced (**Fig. 6c**).

My main concern is the model. The data from the authors strongly suggest that AXL is downstream of *PI3KR2* and not upstream as shown in the final model. All the experiments have been performed in serum, without any AXL-specific stimulation and the phosphorylation of AXL observed by the authors could be also due to cross-phosphorylation by other RTKs. What happens if AXL is stimulated by GAS6? What is the role of *PI3KR2* in this context? This is a big limitation and should be extensively addressed.

Response: We appreciate the insightful comments from the Reviewer and have addressed these interesting questions extensively. First, we examined whether AXL phosphorylation would be a result of cross-phosphorylation by EGFR, which has been shown to crosstalk with AXL. Inhibition of EGFR and HER2 did not alter the induced phosphorylation of AXL by p85 β (**Supplementary Fig. 4b; page 5, lines 29-34**), suggesting that the activation of AXL is unlikely to occur through an interaction with EGFR or HER2.

Second, our new data collectively suggested that Gas6 had minimal inference with the effect of p85 β on AXL (**page 5, line 34 to page 6, line 8; page 7, lines 30-31 and 34-36**).

(1) The induction of AXL phosphorylation by p85 β could be maintained in the presence of Gas6 (**Fig. 3b**). Indeed, the induced phosphorylation of AXL by p85 β and Gas6 was additive.

(2) Total AXL protein levels remained high in p85 β -overexpressing cells after Gas6 stimulation, compared to control-expressing cells (**Fig. 3b**).

(3) We also examined the potential effect of Gas6 in *PIK3R2* knockdown cells. The decreases in AXL phosphorylation and protein level upon *PIK3R2* depletion were not affected by Gas6 (**Fig. 3c**).

(4) The protein interaction between p85 β and AXL remained essentially unchanged in Gas6-stimulated cells (**Fig. 3d**).

(5) The protein interactions between AXL and TRIM2 or optineurin, which were enhanced upon depletion of p85 β , were not altered by Gas6 (**Supplementary Fig. 6c and d**).

The roles of Gas6 and p85 β in AXL are now discussed (**page 12, lines 3-9**).

Also, no protein-protein interactions have been shown in this paper. Does AXL interact with PI3KR2, Trim2 and Optn? This could be tested by proximity ligation assay and co-IP. Does AXL activation by GAS6 affect these interactions? I believe that a deeper mechanistic insight could strengthen this paper and make it suitable for publication.

Response: We apologize that the data of protein-protein interaction were not explained more extensively in the original manuscript. We indeed performed immunoprecipitation experiments for binding between AXL and TRIM2 or optineurin. The data was presented in the original manuscript as Supplementary Fig. 4g (now **Fig. 5e**), which showed that the binding between AXL and TRIM2 or optineurin was enhanced upon *PIK3R2* depletion. In response to this comment, we have also performed proximity ligation assay (PLA) as a supplementary methodology. The PLA yielded consistent results with immunoprecipitation (new data in **Supplementary Fig. 6c and 6d**). As also mentioned above, Gas6 had no influence on these interactions (**page 7, lines 28-31 and lines 34-36**).

Minor comments:

- please indicate in the text where the western blots showing the depletion of PI3KR2 by siRNA or shRNA are located as they are not in figure 1 and suppl figure 1.

Response: Thanks for the reminder. The Western blots showing the depletion of *PIK3R2* are now presented in a separated figure (**Supplementary Fig. 1c**) which is indicated in the text (**page 4, line 12**).

-please show the graphs of the correlation between PI3KR2 and AXL as only the stats are mentioned in the text.

Response: A graph showing the correlation between p85 β and AXL proteins in the tumor samples is now shown in **Fig. 2b (page 5, line 8)**.

[REDACTED]

[REDACTED]

REVIEWERS' COMMENTS:

Reviewer #1 (Remarks to the Author):

Thank you for the revision, I have no other concerns.

Reviewer #2 (Remarks to the Author):

As the authors have adequately addressed my questions in this revised manuscript , I would highly recommend acceptance of it for publication in N.C. asap.

Reviewer #3 (Remarks to the Author):

The authors of the manuscript have worked hard to address all the concerns and the manuscript is now even more solid and convincing, well done! I am very satisfied with the responses and I recommend the publication of this work in Nature Communications